# PF3plat: Pose-Free Feed-Forward 3D Gaussian Splatting

## Abstract

We consider the problem of novel view synthesis from unposed images in a single feed-forward. Our framework capitalizes on fast speed, scalability, and high-quality 3D reconstruction and view synthesis capabilities of 3DGS, where we further extend it to offer a practical solution that relaxes common assumptions such as dense image views, accurate camera poses, and substantial image overlaps. We achieve this through identifying and addressing unique challenges arising from the use of pixel-aligned 3DGS: misaligned 3D Gaussians across different views induce noisy or sparse gradients that destabilize training and hinder convergence, especially when above assumptions are not met. To mitigate this, we employ pre-trained monocular depth estimation and visual correspondence models to achieve coarse alignments of 3D Gaussians. We then introduce lightweight, learnable modules to refine depth and pose estimates from the coarse alignments, improving the quality of 3D reconstruction and novel view synthesis. Furthermore, the refined estimates are leveraged to estimate geometry confidence scores, which assess the reliability of 3D Gaussian centers and condition the prediction of Gaussian parameters accordingly. Extensive evaluations on large-scale real-world datasets demonstrate that PF3plat sets a new state-of-the-art across all benchmarks, supported by comprehensive ablation studies validating our design choices.

## 1 Introduction

In recent years, 3D reconstruction and novel view synthesis have garnered significant attention, particularly with the emergence of Neural Radiance Fields (NeRF) (Mildenhall et al., 2021) and 3D Gaussian Splatting (3DGS) (Kerbl et al., 2023). These advancements have enabled the high-quality 3D reconstruction and novel view synthesis. However, many existing methods rely on stringent assumptions, such as dense image views (Yu et al., 2024; Barron et al., 2021; 2022), accurate camera poses (Kim et al., 2022; Kwak et al., 2023; Zhu et al., 2023; Charatan et al., 2023; Chen et al., 2024), and substantial image overlaps (Yu et al., 2021; Johari et al., 2022), which limit their practical applicability. In real-world scenarios, casually captured images contain sparse and distant viewpoints, and lack precise camera poses, making it impractical to assume densely captured views with accurate camera poses. Ideally, a practical novel view synthesis solution should operate quickly and effectively with as few as two images, even under significant viewpoint changes.

To address some of these limitations, recent efforts (Yu et al., 2021; Johari et al., 2022; Chen et al., 2021; Yang et al., 2023) have introduced generalized view synthesis frameworks capable of performing single feed-forward novel view synthesis from sparse images with minimal overlaps (Du et al., 2023; Xu et al., 2023a). Among these methods, particularly those utilizing 3DGS (Charatan et al., 2023; Chen et al., 2024), have demonstrated remarkable rendering speed and efficiency, alongside impressive reconstruction and view synthesis quality, highlighting the potential of 3D Gaussian-based representations. However, they still depend on accurate camera poses, which are challenging to acquire in sparse settings, thereby restricting their practical use.

More recently, pose-free generalized view synthesis frameworks (Chen & Lee, 2023; Fan et al., 2023; Jiang et al., 2023; Smith et al., 2023; Hong et al., 2024) have been introduced to decouple 3D reconstruction and novel view synthesis from camera poses. Given a set of unposed images, these frameworks aim to jointly learn radiance fields and 3D geometry without relying on addi-

tional data, such as ground-truth camera pose. The learned radiance fields and geometry can then be inferred through trained neural networks, enabling single feed-forward inference. While these pioneering efforts enhance practicality, their performance remains unsatisfactory and their slow rendering speeds (Chen & Lee, 2023; Fan et al., 2023; Jiang et al., 2023; Smith et al., 2023) remain unresolved. In an attempt to boost performance and robustness under extreme scenarios when wide-baseline images are given, Hong et al. (2024) developed a unified framework that jointly estimates camera poses, correspondences, and radiance fields using additional data, such as ground truth poses, for supervision. However, this approach still inherits the inherent limitations of NeRF, including intensive memory consumption and slow rendering speeds, making it impractical for real-world applications.

In this work, we propose **PF3plat** (Pose-Free Feed-Forward 3D Gaussian Splatting), a novel framework for fast and photorealistic novel view synthesis from unposed images in a single feed-forward pass. Our approach leverages the efficiency and high-quality reconstruction capabilities of pixel-aligned 3DGS (Charatan et al., 2023; Szymanowicz et al., 2024), while relaxing common assumptions such as dense image views, accurate camera poses, scene-specific optimization and substantial image overlaps. However, a primary challenge in using pixel-aligned 3DGS is its dependency on precise depth and camera pose estimates for accurate localization of 3D Gaussian centers. Inaccuracies in these estimates can cause misalignments, leading to noisy or sparse gradients that destabilize training and hinder convergence, especially when the above assumptions are relaxed or scene-specific optimizations to rectify errors cannot be applied during multi-scene training.

To mitigate these issues, we find that leveraging pre-trained monocular depth estimation (Piccinelli et al., 2024) and visual correspondence (Lindenberger et al., 2023) models to achieve a coarse alignment of 3D Gaussians is highly effective, thereby promoting a stable learning process. Subsequently, we introduce learnable modules designed to refine the depth and pose estimates from the coarse alignment to enhance the quality of 3D reconstruction and view synthesis. These modules are geometry-aware and lightweight, since we leverage features from the depth network and avoid direct fine-tuning. These refined depth and pose estimates are then used to implement geometry-aware confidence scores to assess the reliability of 3D Gaussian centers, conditioning the prediction of Gaussian parameters such as opacity, covariance, and color.

Our extensive evaluations on large-scale real-world indoor and outdoor datasets (Liu et al., 2021; Zhou et al., 2018; Ling et al., 2024) demonstrate that PF3plat sets a new state-of-the-art across all benchmarks. Comprehensive ablation studies validate our design choices, confirming that our framework provides a fast and high-performance solution for pose-free generalizable novel view synthesis. We summarize our contributions below:

- We address the challenging task of pose-free, feed-forward novel view synthesis using 3DGS, relaxing many common assumptions of existing methods to improve practicality.

- To address the unique challenges arising from the misalignment of 3D Gaussians that destabilizes the learning process, we devise an approach to provide coarse alignment. We then introduce lightweight refinement modules and geometry-aware scoring functions, which not only enhance the reconstruction and view synthesis quality, but also prevent catastrophic forgetting issues typically associated with direct fine-tuning.

- Our framework presents an effective approach to enable fast and high-performance 3D reconstruction and novel view synthesis from sparse and unposed images. We have shown that our method sets a new state-of-the-art across all benchmarks.

## 2 RELATED WORK

**Generalizable Scene Reconstruction and View Synthesis from Unposed Imagery.** Several innovative efforts have addressed the joint learning of camera pose and radiance fields within NeRF-based frameworks. Starting with BARF (Lin et al., 2021), subsequent research (Jeong et al., 2021; Wang et al., 2021; Bian et al., 2023; Truong et al., 2023b) has expanded upon this foundation. Notably, the use of 3D Gaussians as dynamic scene representations has led to significant advancements: Fu et al. (2023) progressively enlarges 3D Gaussians by learning transformations between consecutive frames, SplaTAM(Keetha et al., 2024) utilizes RGB-D sequences and silhouette masks to jointly update Gaussian parameters and camera poses, and InstantSplat (Fan et al., 2024) opti-

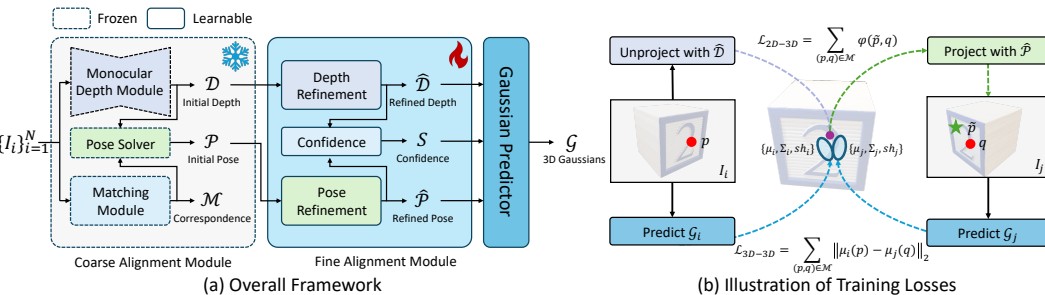

Figure 1: **Overall architecture and loss of the proposed method.** (a) Given a set of unposed images, our method aligns the 3D Gaussians using a coarse-to-fine strategy. (b) In addition to photometric loss, we enforce 3D Gaussian consistency by ensuring they are placed on the same object surface through 2D-3D and 3D-3D consistency losses.

mizes 3D Gaussians rapidly for scene reconstruction and view synthesis. Among these, methods like DBARF (Chen & Lee, 2023), FlowCAM (Smith et al., 2023), CoPoNeRF (Hong et al., 2023), and GGRt (Li et al., 2024) aim to determine camera pose and radiance fields in a single feed-forward pass. Concurrently, Splatt3R (Smart et al., 2024) builds on pre-trained 3D reconstruction models (Leroy et al., 2024) to relax some assumptions; however, it still relies on additional data such as ground truth depth and pose. In contrast, our method reconstructs 3D scenes and synthesizes novel views from unposed images with minimal overlaps in a single feed-forward pass, eliminating the need for scene-specific optimization and extensive pose data.

**Monocular Depth Estimation and Correspondence Estimation.** Monocular depth estimation and visual correspondence estimation are fundamental computer vision tasks with extensive research spanning decades. Recent advancements (Yin et al., 2023; Piccinelli et al., 2024; Ke et al., 2024; Yang et al., 2024) in monocular depth estimation have matured these models, significantly enhancing various 3D vision applications. Similarly, visual correspondence estimation has advanced significantly since the emergence of deep neural networks. Conventionally, the task is evaluated primarily in camera pose estimation, where classical methods follow a pipeline that includes keypoint detection and descriptor extraction (Lowe, 2004; Bay et al., 2006; Tola et al., 2008; Salti et al., 2014; Rusu et al., 2009), tentative matching and outlier filtering (Fischler & Bolles, 1981), and camera pose estimation using solvers (Gower, 1975; Nistér, 2004; Hartley, 1997; Hesch & Roumeliotis, 2011). More recently, deep learning-based approaches have optimized each stage, outperforming traditional methods in tasks such as 2D descriptor extraction (Yi et al., 2016; DeTone et al., 2018), 3D descriptor extraction (Yew & Lee, 2018; Choy et al., 2019), sparse and dense matching (Hong & Kim, 2021; Cho et al., 2021; 2022; Hong et al., 2022a;b; 2023; Sun et al., 2021; Edstedt et al., 2024), and outlier filtering (Barath et al., 2019; Wei et al., 2023). In this work, we propose a lightweight plug-and-play depth and pose refinement modules to enhance the quality of view synthesis.

## 3 METHOD

### 3.1 PROBLEM FORMULATION

Our objective is to reconstruct a 3D scene from a set of $N$ unposed images $\{I_i\}_{i=1}^N$ with $I_i \in \mathbb{R}^{H \times W \times 3}$ and synthesize photo-realistic images $\hat{I}_t$ from novel viewpoints in a single feed-forward pass. To achieve this, we output the depth maps $\mathcal{D}_i \in \mathbb{R}^{H \times W}$ for each image $I_i$, along with their corresponding camera poses $\mathcal{P}_i \in \mathbb{R}^{3 \times 4}$, consisting of a rotation matrix $R_i \in \mathbb{R}^{3 \times 3}$ and a translation vector $t_i \in \mathbb{R}^{3 \times 1}$. Additionally, we compute a set of pixel-aligned 3D Gaussians denoted as $\mathcal{G} = \{\mu_i, \sigma_i, \Sigma_i, c_i\}_{i=1}^N$. Here, $\mu_i(p) \in \mathbb{R}^3$ indicates the 3D Gaussian center derived from the depth $\mathcal{D}_i(p)$, camera pose $\mathcal{P}_i$, and camera intrinsic $K_i$, where $p \in \mathbb{R}^{H \times W}$ represents each pixel. The opacity is represented by $\sigma_i(p) \in [0, 1)$, $\Sigma_i(p) \in \mathbb{R}^{3 \times 3}$ is the covariance matrix, and the color is encoded using spherical harmonics $c_i(p) \in \mathbb{R}^{3(L+1)}$, where $L$ is the order of the spherical harmonics. Note that, for both training and inference, we *do not use any ground truth camera poses*. Instead, we let the network predict the camera poses for all input images while also learning the varying scale factors across different scenes.

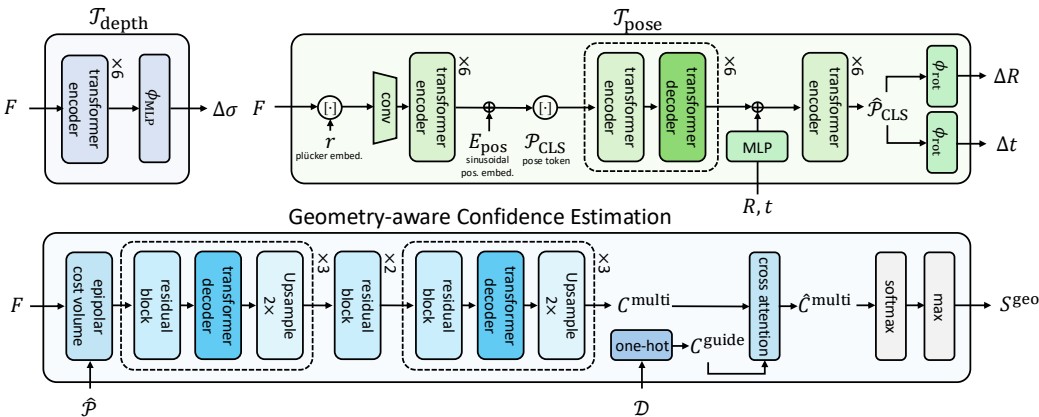

Figure 2: **Proposed refinement and confidence estimation modules.** In our Fine Alignment module, we refine depth and pose to improve 3D reconstruction and view synthesis quality, alongside estimating confidence to assess the reliability of predicted 3D Gaussian centers.

## 3.2 PF3PLAT: POSE-FREE FEED-FORWARD 3D GAUSSIAN SPLATTING

### 3.2.1 COARSE ALIGNMENT OF 3D GAUSSIANS

Inspired by recent advancements (Charatan et al., 2023; Chen et al., 2024) that highlight the advantages of pixel-aligned 3D Gaussians, such as speed, efficiency, and high-quality reconstruction and view synthesis, we extend these benefits to more challenging scenarios, specifically in the context of pose-free feed-forward view synthesis. To this end, we adopt pixel-aligned 3D Gaussians as our scene representation. However, this representation also poses certain challenges. Unlike previous methods for generalized novel view synthesis that utilize implicit representations (Chen & Lee, 2023; Smith et al., 2023; Hong et al., 2024) and benefit from the interpolation capabilities of neural networks, our approach is challenged by the explicit nature of this representation. Specifically, our method directly localizes 3D Gaussian centers using depth and camera pose estimates (Charatan et al., 2023), making it highly sensitive to inaccuracies in these estimates, which cannot be easily compensated.

Such misalignments can cause severe performance degradation and disrupt the learning process by producing sparse and noisy gradients. This issue is particularly exacerbated when wide-baseline images are given as input or the absence of ground-truth pose prevents alignments of 3D Gaussians. Without effectively addressing these challenges, we find the problem becomes nearly intractable. These issues can be mitigated by employing iterative scene-specific optimization steps or by assuming ground-truth camera poses to guide 3D Gaussians toward object surfaces. However, these solutions are incompatible with our goal of achieving a single feed-forward process from unposed images. Therefore, overcoming these limitations requires a novel strategy that can handle depth and pose ambiguities while maintaining efficiency in a feed-forward manner.

To mitigate the challenges mentioned above, we find it **necessary** to provide coarse alignment of 3D Gaussians, where quantitative results can be found in Tab. 4. To this end, we employ off-the-shelf models (Piccinelli et al., 2024; Lindenberger et al., 2023) to estimate initial depths $\mathcal{D}_i$ and camera poses $\mathcal{P}_i$ for our images $I_i$. Specifically, given depth maps $\mathcal{D}_i$ and sets of correspondences $\mathcal{M}_{ij}$ and their confidence values $\mathcal{C}_{ij}$ acquired from each pairwise combinations of images, *e.g.*, $(I_i, I_j)$, where $i, j$ refer to image indices, we use a robust solver (Fischler & Bolles, 1981; Li et al., 2012) to estimate the relative poses $\mathcal{P}_{ij}$ between image pair. Integrating these components, we provide the necessary coarse alignment to promote stabilizing the learning process and serve as a strong foundation for further enhancements.

### 3.2.2 MULTI-VIEW CONSISTENT DEPTH ESTIMATION

While pre-trained monocular depth models (Wang et al., 2023; Leroy et al., 2024; Piccinelli et al., 2024; Yin et al., 2023) can offer powerful 3D geometry priors, inherent limitations of these models, namely, inconsistent scales among predictions, still remain unaddressed. This requires further ad-

justments to ensure multiview consistency across predictions. To overcome this challenge, we aim to refine the predicted depths and camera poses obtained from coarse alignment in a fully learnable and differentiable manner.

Our refinement module includes a pixel-wise depth offset estimation that uses the feature maps $F_i$ from the depth network (Piccinelli et al., 2024) as the sole input and processes them through a series of self-attention operations, making it lightweight and geometry-aware (Xu et al., 2023b). The process is defined as:

$$\Delta\delta_i = \phi_{\mathrm{mlp}}(\mathcal{T}_{\mathrm{depth}}(F_i)),$$
$$\hat{\mathcal{D}}_i = \mathcal{D}_i + \Delta\delta_i, \tag{1}$$

where $\phi_{\mathrm{mlp}}(\cdot)$ is a linear projection, $\mathcal{T}_{\mathrm{depth}}$ is a deep Transformer architecture and $\Delta\delta$ is the pixel-wise depth offset. This extension promotes consistency across views and enhances performance without relying on explicit cross-attention. Instead, it leverages supervision signals derived from pixel-aligned 3D Gaussians that connect the information across views, and leverage them for the novel view synthesis task (Zhou et al., 2017) and our loss functions, which are detailed in Section 3.3. Additionally, we avoid fine-tuning the entire depth network, thereby reducing computational costs and mitigating the risk of catastrophic forgetting.

### 3.2.3 CAMERA POSE REFINEMENT

In this module, we further refine camera poses to enhance reconstruction and view synthesis quality. Initially, we replace the estimated relative poses $\mathcal{P}_{ij}$ with newly computed camera poses $\hat{\mathcal{P}}_{ij}$ derived from following the similar process in coarse alignment and using the previously obtained refined depths $\hat{\mathcal{D}}_i$. We then introduce a learnable camera pose refinement module that estimates rotation and translation offsets. To streamline this process, we first utilize a fully differentiable transformation synchronization operation that takes $\hat{\mathcal{P}}_{ij}$ and $\mathcal{C}_{ij}$ as inputs. Using power iterations (El Banani et al., 2023), this operation efficiently recovers the absolute poses $\hat{\mathcal{P}}_i$ prior to the refinement module.

Next, we convert the absolute poses $\hat{\mathcal{P}}_i$ into Plücker coordinates (Sitzmann et al., 2021), defined as $r = (\mathrm{d}, \mathrm{o}\times\mathrm{d}) \in \mathbb{R}^6$, where d represents the camera direction and o denotes the camera origin. These coordinates, along with the feature maps $F_i \in \mathbb{R}^{h\times w\times d}$ and a pose token $\mathcal{P}_{\mathrm{CLS}} \in \mathbb{R}^d$, are input into a series of self- and cross-attention layers. In our approach, we designate $\hat{\mathcal{P}}_1$ as the reference world space and update only the other pose estimates. The resulting pose token is then transformed into 6D rotations (Zhou et al., 2019) and translations, which are added to the initial camera poses to estimate the rotation and translation offsets. Formally, these are defined as:

$$\hat{\mathcal{P}}_{\mathrm{CLS}} = \mathcal{T}_{\mathrm{pose}}([F_i, \mathcal{P}_{\mathrm{CLS}}, r] + E_{\mathrm{pos}}),$$
$$\Delta R_i, \Delta t_i = \phi_{\mathrm{rot}}(\hat{\mathcal{P}}_{\mathrm{CLS}}), \phi_{\mathrm{trans}}(\hat{\mathcal{P}}_{\mathrm{CLS}}), \tag{2}$$
$$\hat{R}_i, \hat{t}_i \Leftarrow \hat{R}_i + \Delta R_i, \hat{t} + \Delta t,$$

where $\Delta R, \Delta t$ are the pose offsets, and $E_{\mathrm{pos}})$ is positional embedding.

### 3.2.4 3D GAUSSIAN PARAMTER PREDICTIONS

**Cost Volume Construction and Aggregation.** Using the refined pose and the monocular depth estimates, $\hat{\mathcal{P}}_i$ and $\hat{\mathcal{D}}_i$, we assess the quality of the predictions to obtain confidence scores, to assist predicting 3D Gaussian parameters. To achieve this, we construct both a conventional multi-view stereo cost volume and a guidance cost volume derived from $\mathcal{D}_i$.

Specifically, given the current pose estimates $\hat{\mathcal{P}}_i$, we build a multi-view stereo cost volume $\mathcal{C}_i^{\mathrm{multi}} \in \mathbb{R}^{h\times w\times K}$ following the plane-sweeping approach (Yao et al., 2018; Chen et al., 2021). For each of the $K$ depth candidates within specified near and far ranges along the epipolar lines, we compute matching scores using cosine similarity (Cho et al., 2022; 2021; Hong et al., 2024). Subsequently, to guide the depth localization along the epipolar lines, we construct a guidance cost volume $\mathcal{C}_i^{\mathrm{guide}}$ (Li et al., 2023), where each spatial location is represented by a one-hot vector indicating the depth candidate closest to the monocular depth estimate. The constructed $\mathcal{C}_i^{\mathrm{multi}}$ and $\mathcal{C}_i^{\mathrm{guide}}$ undergo cross-attention to update the multi-view cost volume $\hat{\mathcal{C}}_i^{\mathrm{multi}}$.

**Geometry-aware Confidence Estimation.**    From the updated multi-view cost volume $\hat{\mathcal{C}}_i^{\mathrm{multi}}$, we apply a softmax function (Xu et al., 2022; Hong et al., 2023) along the $K$ dimension to obtain a matching distribution. We then extract the maximum value from this distribution to derive a confidence score, $S_{\mathrm{geo}}$, which assesses the quality of the estimated camera pose and depth. Formally, this is defined as:

$$S_i^{\mathrm{geo}} = \max_{k \in \{1,2,...,K\}} \mathrm{softmax}(\hat{\mathcal{C}}_i^{\mathrm{multi}})(k). \tag{3}$$

These confidence scores assess the reliability of the predicted 3D Gaussian centers, where high confidence indicates accurate localization and low confidence suggests potential inaccuracies due to noise or misalignment. To condition the prediction of Gaussian parameters, opacity, covariance, and color, we incorporate $S_{\mathrm{geo}}$ as additional input.

**3D Gaussian Parameters.**    Finally, using the inputs $[I_i, \hat{\mathcal{D}}_i, F_i, S_i^{\mathrm{geo}}]$, we compute the opacity $\sigma_i$ through small convolutional layers, derive the covariances from the estimated rotations and scales, and obtain the color from the estimated spherical harmonic (SH) coefficients. A key idea of our approach is that $S^{\mathrm{geo}}$ enables supervision signals to flow from the Gaussian parameters back to the depth and pose estimates. This feedback loop enhances the accuracy of both depth and pose estimations, resulting in more consistent and reliable 3D reconstructions.

## 3.3 LOSS FUNCTION

**Reconstruction Loss.**    By making both depth and camera pose refinement modules learnable and differentiable, our network leverages supervision signals from the pixel-aligned 3D Gaussians and the novel view synthesis task (Zhou et al., 2017) to refine these estimates during training. Specifically, we combine photometric loss, defined as the L2 loss between the rendered and target images, as well as SSIM (Wang et al., 2004) loss $\mathcal{L}_{\mathrm{SSIM}}$ and LPIPS (Zhang et al., 2018) loss $\mathcal{L}_{\mathrm{LPIPS}}$ to form our reconstruction loss $\mathcal{L}_{\mathrm{img}}$.

**2D-3D Consistency Loss.**    We identify that provided good coarse alignments, RGB loss is sufficient, but with larger baselines, the training process starts to destabilize. Moreover, one remaining issue with learning solely from the photometric loss is that the gradients are mainly derived from pixel intensity differences, which suffer in textureless regions. To remedy these, we enforce that corresponding points in the set of images $\{I\}_{i=1}^N$ lie on the same object surface, drawing from principles of multi-view geometry Hartley & Zisserman (2003).

Formally, using the estimated depths $\hat{D}$, the camera poses $\hat{\mathcal{P}}$, and the correspondence sets $\mathcal{M}$, we can define a geometric consistency loss that penalizes deviations from the multi-view geometric constraints. For each correspondence $(p, q) \in \mathcal{M}_{ij}$ between images $I_i$ and $I_j$, we compute the 3D point from the pixel $p$ and its estimated depth $\hat{D}_i(p)$ using the camera intrinsics. We then transform this to the coordinate frame of $I_j$ using the relative pose $\hat{\mathcal{P}}_{ij}$ and project it back onto the image plane to obtain the predicted correspondence $\tilde{p}$. This is defined as $\mathcal{L}_{\mathrm{2D-3D}} = \sum_{(p,q) \in \mathcal{M}} \varphi(\tilde{p}, q)$, where $\varphi(\cdot)$ denotes huber loss. By integrating this loss into the learning pipeline, we improve the robustness of the model in regions with low texture or significant viewpoint changes, ensuring that the estimated pose and depth are consistent with the underlying 3D structure of the scene.

**3D-3D Consistency Loss.**    While the multi-view consistent surface loss directly connects each pair of corresponding Gaussians and their centers, guiding the model towards the object's surface, we find that relying solely on this loss can lead to suboptimal convergence, especially in regions with sparse correspondences. To further stabilize and enhance the learning process, we introduce an additional regularization term that minimizes the discrepancies among the centers of the corresponding Gaussians.

Intuitively, this differs from $\mathcal{L}_{\mathrm{2D-3D}}$ in that, unlike the previous function, which considers the alignment of Gaussian centers from only one side when dealing with pairwise correspondences, the regularization term symmetrically enforces consistency from both sides. Specifically, while the multi-view consistent surface loss projects the Gaussian center from one view to another using the estimated depth and camera pose, *e.g.,* from source to target, the regularization term jointly minimizes the distances between all corresponding Gaussian centers across multiple views. By considering both directions in pairwise correspondences, this approach promotes a more coherent and robust estimation of the object's surface, reducing the influence of outliers and im-

| | | RealEstate-10K | | | | | | | | | | | | | | | |
|---|---|---|---|---|---|---|---|---|---|---|---|---|---|---|---|---|---|
| | | Avg | | | | Small | | | | Medium | | | | Large | | | |
| Pose-Free | Method | PSNR | LPIPS | SSIM | MSE | PSNR | LPIPS | SSIM | MSE | PSNR | LPIPS | SSIM | MSE | PSNR | LPIPS | SSIM | MSE |
| ✗ | PixelNeRF | 14.438 | 0.577 | 0.467 | 0.047 | 13.126 | 0.639 | 0.466 | 0.058 | 13.999 | 0.582 | 0.462 | 0.042 | 15.448 | 0.479 | 0.470 | 0.031 |
| | Du et al. | 21.833 | 0.294 | 0.736 | 0.011 | 18.733 | 0.378 | 0.661 | 0.018 | 22.552 | 0.263 | 0.764 | 0.008 | 26.199 | 0.182 | 0.836 | 0.004 |
| | MVSplat | 25.054 | 0.157 | 0.827 | 0.008 | 21.029 | 0.226 | 0.747 | 0.013 | 26.369 | 0.116 | 0.874 | 0.004 | 30.516 | 0.074 | 0.926 | 0.002 |
| ✓ | DBARF | 14.789 | 0.490 | 0.570 | 0.033 | 13.453 | 0.563 | 0.522 | 0.045 | 15.201 | 0.487 | 0.560 | 0.030 | 16.615 | 0.380 | 0.648 | 0.022 |
| | FlowCAM | 18.242 | 0.597 | 0.455 | 0.023 | 15.435 | 0.528 | 0.570 | 0.034 | 18.481 | 0.592 | 0.441 | 0.18 | 22.418 | 0.707 | 0.287 | 0.009 |
| | CoPoNeRF | 19.536 | 0.398 | 0.638 | 0.016 | 17.153 | 0.459 | 0.577 | 0.025 | 19.965 | 0.343 | 0.645 | 0.013 | 22.542 | 0.250 | 0.724 | 0.008 |
| | Ours | **22.347** | **0.205** | **0.763** | **0.010** | **18.904** | **0.276** | **0.683** | **0.017** | **23.366** | **0.173** | **0.793** | **0.007** | **27.064** | **0.115** | **0.870** | **0.003** |

| | | ACID | | | | | | | | | | | | | | | |
|---|---|---|---|---|---|---|---|---|---|---|---|---|---|---|---|---|---|
| | | Avg | | | | Small | | | | Medium | | | | Large | | | |
| Pose-Free | Method | PSNR | LPIPS | SSIM | MSE | PSNR | LPIPS | SSIM | MSE | PSNR | LPIPS | SSIM | MSE | PSNR | LPIPS | SSIM | MSE |
| ✗ | PixelNeRF | 17.160 | 0.527 | 0.500 | 0.029 | 16.996 | 0.528 | 0.487 | 0.030 | 17.228 | 0.534 | 0.501 | 0.029 | 17.229 | 0.522 | 0.500 | 0.028 |
| | Du et al. | 25.482 | 0.304 | 0.769 | 0.005 | 25.553 | 0.301 | 0.773 | 0.005 | 25.694 | 0.303 | 0.769 | 0.005 | 25.338 | 0.307 | 0.763 | 0.005 |
| | MVSplat | 28.252 | 0.157 | 0.829 | 0.004 | 28.085 | 0.164 | 0.820 | 0.004 | 28.571 | 0.148 | 0.843 | 0.003 | 28.203 | 0.156 | 0.828 | 0.004 |
| ✓ | DBARF | 14.189 | 0.452 | 0.537 | 0.038 | 14.306 | 0.503 | 0.541 | 0.037 | 14.253 | 0.457 | 0.538 | 0.038 | 14.086 | 0.419 | 0.534 | 0.039 |
| | FlowCAM | 20.116 | 0.477 | 0.585 | 0.016 | 20.153 | 0.475 | 0.594 | 0.016 | 20.158 | 0.476 | 0.585 | 0.015 | 20.073 | 0.478 | 0.580 | 0.016 |
| | CoPoNeRF | 22.440 | 0.323 | 0.649 | 0.010 | 22.322 | 0.358 | 0.649 | 0.010 | 22.407 | 0.352 | 0.648 | 0.009 | 22.529 | 0.351 | 0.649 | 0.009 |
| | Ours | **23.732** | **0.251** | **0.702** | **0.008** | **23.719** | **0.250** | **0.702** | **0.009** | **23.935** | **0.246** | **0.708** | **0.008** | **23.647** | **0.253** | **0.695** | **0.008** |

Table 1: **Novel View Synthesis Performance on RealEstate-10K and ACID.** Gray entries indicate methods that use ground truth camera poses during evaluation and are not directly comparable.

proving convergence during training. This additional regularization can be formally defined as: $\mathcal{L}_{3D-3D} = \sum_{(p,q)\in\mathcal{M}} ||\mu_i(p) - \mu_j(q)||_2$.

**Final Objective Function.** Combining the three loss functions, we define our final objective function: $\mathcal{L}_{img} + \mathcal{L}_{2D-3D} + \lambda_{3D-3D}\mathcal{L}_{3D-3D}$, where we set $\lambda_{3D-3D} = 0.05$

# 4 EXPERIMENTS

## 4.1 IMPLEMENTATION DETAILS

In this work, we assume intrinsic parameters are given, as they are generally available from modern devices (Arnold et al., 2022). We compute attentions using Flash Attention (Dao et al., 2022), and for the Gaussian rasterizer, we follow the method described in (Charatan et al., 2023). Our model is trained on a single NVIDIA A6000 GPU for 40,000 iterations using the Adam optimizer (Kingma, 2014), with a learning rate set to $2 \times 10^{-4}$ and a batch size of 5, which takes approximately two days. For training on the RealEstate10K and ACID datasets, we gradually increase the number of frames between $I_1$ and $I_2$ as training progresses, initially setting the frame distance to 45 and gradually increasing it to 75. For the DL3DV dataset, we start with a frame distance of 5 and increase it to 10. The target view is randomly sampled within this range. We train under the assumption of $N = 2$ and render $\hat{I}_t$. Our code is implemented using PyTorch (Paszke et al., 2017). Additional implementation details can be found in the supplementary material. The code and pretrained weights will be made publicly available.

## 4.2 EXPERIMENTAL SETTING

**Datasets.** We train and evaluate our method on three large-scale datasets: RealEstate10K (Zhou et al., 2018), a collection of both indoor and outdoor scenes; ACID (Liu et al., 2021), a dataset focusing on outdoor coastal scenes; and DL3DV (Ling et al., 2024), which includes diverse real-world indoor and outdoor environments. For RealEstate10K, due to some unavailable videos on YouTube, we use a subset of the full dataset, comprising a training set of 21,618 scenes and a test set of 7,200 scenes. For ACID, we train on 10,935 scenes and evaluate on 1,893 scenes. Lastly, for DL3DV, which features longer video sequences across 10,510 different scenes, we use 2,000 scenes for training and a standard benchmark set of 140 scenes for testing (Ling et al., 2024).

**Baselines.** Following Hong et al. (2024), we evaluate our method on two tasks: novel-view synthesis and camera pose estimation. For novel view synthesis, we compare our approach against established generalized NeRF and 3DGS variants, including PixelNeRF (Yu et al., 2021), Du et al. (2023), PixelSplat (Charatan et al., 2023), and MVSPlat (Chen et al., 2024). It is important to note that these methods assume ground-truth (GT) camera poses during inference, so we include them only for reference. Our primary comparisons focus on existing pose-free generalized novel view

| Task | Method | RealEstate-10K | | | | | | | | | | | | | | | |
|---|---|---|---|---|---|---|---|---|---|---|---|---|---|---|---|---|---|
| | | Avg | | | | Small | | | | Medium | | | | Large | | | |
| | | Rotation | | Translation | | Rotation | | Translation | | Rotation | | Translation | | Rotation | | Translation | |
| | | Avg(°↓) | Med(°↓) | Avg(°↓) | Med(°↓) | Avg(°↓) | Med(°↓) | Avg(°↓) | Med(°↓) | Avg(°↓) | Med(°↓) | Avg(°↓) | Med(°↓) | Avg(°↓) | Med(°↓) | Avg(°↓) | Med(°↓) |
| SfM | SP+SG | 5.605 | 1.301 | 14.89 | 5.058 | 9.793 | 2.270 | 12.55 | 4.638 | 1.789 | 0.969 | 9.295 | 3.279 | 1.416 | 0.847 | 21.42 | 7.190 |
| | PDC-Net+ | 2.189 | 0.751 | 10.10 | 3.243 | 3.460 | 1.128 | 6.913 | 2.752 | 1.038 | 0.607 | 6.667 | 2.262 | 0.981 | 0.533 | 16.57 | 5.447 |
| | DUSt3R | 2.527 | 0.814 | 17.45 | 4.131 | 3.856 | 1.157 | 12.23 | 2.899 | 1.650 | 0.733 | 14.00 | 3.650 | 0.957 | 0.476 | 27.30 | 10.27 |
| | MASt3R | 2.555 | 0.751 | 9.775 | 2.830 | 4.240 | 1.283 | 8.050 | 2.515 | 1.037 | 0.573 | 6.904 | 2.418 | 0.791 | 0.418 | 13.963 | 3.925 |
| Pose Estimation | 8ViT | 12.59 | 6.881 | 90.12 | 88.65 | 12.60 | 6.860 | 91.46 | 91.50 | 12.17 | 6.552 | 82.48 | 82.92 | 12.77 | 7.214 | 91.85 | 88.92 |
| | RelPose | 8.285 | 3.845 | - | - | 12.10 | 4.803 | - | - | 4.942 | 3.476 | - | - | 4.217 | 2.447 | - | - |
| Pose-Free View Synthesis | DBARF | 11.14 | 5.385 | 93.30 | 102.5 | 17.52 | 13.22 | 126.3 | 140.4 | 7.254 | 4.379 | 79.40 | 75.41 | 3.455 | 1.937 | 50.09 | 33.96 |
| | FlowCAM | 7.426 | 4.051 | 50.66 | 46.28 | 11.88 | 6.778 | 87.12 | 58.25 | 4.154 | 3.346 | 42.29 | 41.59 | 2.349 | 1.524 | 34.47 | 27.79 |
| | CoPoNeRF | 3.610 | 1.759 | 12.77 | 7.534 | 5.471 | 2.551 | 11.86 | 5.344 | 2.183 | 1.485 | 10.19 | 5.749 | 1.529 | 0.991 | 15.544 | 7.907 |
| | Ours | **1.965** | **0.751** | **10.113** | **4.785** | **2.561** | **1.031** | **7.349** | **4.122** | **1.536** | 1.536 | **9.332** | **4.525** | **1.278** | **0.550** | 14.753 | 6.127 |
| Task | Method | ACID | | | | | | | | | | | | | | | |
| | | Avg | | | | Small | | | | Medium | | | | Large | | | |
| | | Rotation | | Translation | | Rotation | | Translation | | Rotation | | Translation | | Rotation | | Translation | |
| | | Avg(°↓) | Med(°↓) | Avg(°↓) | Med(°↓) | Avg(°↓) | Med(°↓) | Avg(°↓) | Med(°↓) | Avg(°↓) | Med(°↓) | Avg(°↓) | Med(°↓) | Avg(°↓) | Med(°↓) | Avg(°↓) | Med(°↓) |
| SfM | SP+SG | 4.819 | 1.203 | 20.802 | 6.878 | 10.920 | 2.797 | 22.214 | 7.526 | 3.275 | 1.306 | 16.455 | 5.426 | 1.851 | 0.745 | 22.018 | 7.309 |
| | PDC-Net+ | 4.830 | 1.742 | 48.409 | 28.258 | 2.520 | 0.579 | 15.664 | 4.215 | 2.378 | 0.688 | 14.940 | 4.301 | 1.953 | 0.636 | 18.447 | 4.357 |
| | DUSt3R | 5.558 | 1.438 | 50.661 | 36.154 | 6.515 | 1.450 | 51.348 | 39.334 | 4.773 | 1.392 | 49.647 | 35.105 | 5.346 | 1.444 | 50.724 | 35.260 |
| | MASt3R | 2.320 | 0.625 | 25.325 | 7.534 | 2.223 | 0.647 | 25.382 | 8.107 | 1.977 | 0.613 | 24.460 | 6.635 | 2.544 | 0.613 | 25.697 | 7.099 |
| Pose Estimation | 8ViT | 4.568 | 1.312 | 88.433 | 88.961 | 8.466 | 3.151 | 88.421 | 88.958 | 4.325 | 1.564 | 90.555 | 90.799 | 2.280 | 0.699 | 86.580 | 87.559 |
| | RelPose | 6.348 | 2.567 | - | - | 10.081 | 4.753 | - | - | 5.801 | 2.803 | - | - | 4.309 | 2.011 | - | - |
| Pose-Free View Synthesis | DBARF | 4.681 | 1.421 | 71.711 | 68.892 | 8.721 | 3.205 | 95.149 | 99.490 | 4.424 | 1.685 | 77.324 | 77.291 | 2.303 | 0.859 | 54.523 | 38.829 |
| | FlowCAM | 9.001 | 6.749 | 95.405 | 88.133 | 8.663 | 6.675 | 92.130 | 85.846 | 8.778 | 6.589 | 95.444 | 87.308 | 9.305 | 6.898 | 97.392 | 89.359 |
| | CoPoNeRF | 3.283 | 1.134 | 22.809 | 14.502 | 3.548 | 1.129 | 23.689 | 11.289 | 2.573 | 1.169 | 21.401 | 10.656 | 3.455 | 1.129 | 22.935 | 10.588 |
| | Ours | 4.125 | 1.776 | 27.727 | 13.903 | 4.011 | 1.604 | 27.786 | 13.840 | 3.667 | 1.604 | 26.343 | 13.622 | 4.412 | 1.891 | 28.330 | 14.092 |

Table 2: **Pose Estimation Performance on RealEstate-10K and ACID.** Gray entries indicate methods not trained on the same dataset due to the absence of ground truth data (e.g., depth and correspondence), making them incomparable.

| Pose-Free | Method | DL3DV | | | | | | | | | | | | |
|---|---|---|---|---|---|---|---|---|---|---|---|---|---|---|
| | | Small | | | | | | | Large | | | | | |
| | | PSNR | LPIPS | SSIM | Rotation | | Translation | | PSNR | LPIPS | SSIM | Rotation | | Translation | |
| | | | | | Avg. | Med. | Avg. | Med. | | | | Avg. | Med. | Avg. | Med. |
| ✗ | MVSPlat | 20.849 | 0.230 | 0.680 | - | - | - | - | 24.211 | 0.147 | 0.796 | - | - | - | - |
| ✓ | CoPoNeRF | 15.509 | 0.563 | 0.396 | 13.121 | 6.721 | 44.645 | 30.269 | 17.586 | 0.467 | 0.469 | 5.609 | 2.905 | 17.974 | 12.445 |
| | Ours | 19.355 | 0.280 | 0.611 | 4.698 | 2.475 | 10.692 | 6.869 | 22.105 | 0.211 | 0.706 | 3.353 | 1.604 | 9.407 | 6.334 |

Table 3: **Novel View Synthesis and Pose Estimation Performance on DL3DV**. We include MVS-Plat for reference only.

synthesis methods, such as DBARF (Chen & Lee, 2023), FlowCAM (Smith et al., 2023), and Co-PoNeRF (Hong et al., 2024). For camera pose estimation, we first evaluate against correspondence-based pose estimation methods (e.g., COLMAP), including SP+SG (DeTone et al., 2018; Sarlin et al., 2020), PDC-Net+(Truong et al., 2023a), DUSt3R(Wang et al., 2023), and MASt3R (Leroy et al., 2024). Additionally, we compare with direct pose regression methods such as 8ViT (Rockwell et al., 2022), RelPose (Zhang et al., 2022), and MicKey (Barroso-Laguna et al., 2024). Our main comparisons, however, are with existing pose-free approaches, including DBARF (Chen & Lee, 2023), FlowCAM (Smith et al., 2023), and CoPoNeRF (Hong et al., 2024).

**Evaluation Protocol** For the evaluation on RealEstate-10K and ACID, we follow the protocol outlined by Hong et al. (2024), where evaluation is conducted using unposed triplet images $(I_1, I_2, I_t)$. The test set is divided into three groups, small, middle, and large, based on the extent of overlap between $I_1$ and $I_2$. This allows the model's performance to be assessed under varying levels of difficulty, reflecting different real-world scenarios. For the relatively new DL3DV dataset, we introduce a new evaluation protocol. For each scene, we select two context images, $I_1$ and $I_2$, by skipping frames at intervals of 5 and 10, creating two groups per scene, each representing small and large overlap cases. We then randomly select three target images from the sequence between the context images, similar to the above protocol. For evaluation metrics, we use standard image quality measures, PSNR, SSIM, LPIPS, and MSE, for novel view synthesis. For camera pose estimation, we compute the geodesic rotation error and angular difference in translation, as commonly done in classical methods (Nistér, 2004; Melekhov et al., 2017). These errors are measured by comparing the ground truth relative pose $P_{12}^{\text{GT}}$ and our estimated pose $\hat{\mathcal{P}}_{12}$. Our statistical analysis reports both the average and median errors, with the median providing robustness against outliers.

## 4.3 EXPERIMENTAL RESULTS

**RealEstate-10K & ACID.** Tab.1 summarizes the performance for the novel view synthesis task, while Tab.2 reports the results for pose estimation. From the results in Tab. 1, our method significantly outperforms previous pose-free generalizable methods (Chen & Lee, 2023; Smith et al., 2023; Hong et al., 2024), setting a new state-of-the-art across these benchmarks. Furthermore, compared to the previous state-of-the-art, CoPoNeRF, our method achieves a 2.8 dB improvement in PSNR,

demonstrating superior reconstruction quality and robustness. Additionally, our approach demonstrates superior pose estimation performance on RealEstate-10K; however, we observe that Hong et al. (2024) achieves lower pose errors on the ACID dataset. This discrepancy may be attributed to the larger scale of scenes, such as coastal landscapes and sky views, which complicates our refinement process and poses challenges for our depth network in estimating the metric depth of the scene. Nonetheless, this limitation is mitigated by our method's superior novel view image quality and the fact that Hong et al. (2024) utilizes ground-truth poses for supervision, providing robust guidance for large-scale environments. Moreover, the ACID dataset includes numerous dynamic scenes, which are beyond the scope of our current focus. Consequently, our method may be less effective in estimating poses for dynamic scenes compared to other approaches.

**DL3DV.** While RealEstate-10K and ACID encompass a variety of indoor and outdoor scenes, RealEstate-10K predominantly includes indoor environments, whereas ACID features numerous dynamic scenes. To comprehensively evaluate our method across a broader spectrum of real-world scenarios, we further assess it on the recently released DL3DV dataset (Ling et al., 2024). The results are summarized in Table 3. From these results, we observe that our method outperforms CoPoNeRF (Hong et al., 2024) by over 4 dB in large-overlap scenarios and by 3.8 dB in small-overlap scenarios, highlighting the superior accuracy and robustness of our approach in handling diverse and complex environments. This highlights the effectiveness of our method in managing varied scene and object types, reinforcing its applicability for practical novel view synthesis tasks.

## 4.4 ABLATION STUDY

In this ablation study, we aim to investigate the effectiveness of each component of our method. We first define a baseline model, which combines our depth and pose estimates from coarse alignments with MVSplat (Chen et al., 2024) for 3D Gaussian parameter prediction. We also explore both full fine-tuning and partial fine-tuning strategies for the depth network. Additionally, we report the results of ablation studies on our loss functions. The results are summarized in Tab. 4.

From the results, we find that our method improves by a large margin when comparing (**0**) with (**I**). This improvement is further supported by the comparisons from (**I**) to (**IV**) and from (**I-III**) to (**I-IV**), which show performance degradation as each component is removed. We also demonstrate that without pre-trained weights for the depth and correspondence networks, the training either fails or achieves significantly lower

|  | Components | Avg | | | | | | |
|---|---|---|---|---|---|---|---|---|
|  |  | PSNR | SSIM | LPIPS | Rotation | | Translation | |
|  |  |  |  |  | Avg. | Med. | Avg. | Med. |
| **(0)** | Baseline | 20.140 | 0.694 | 0.281 | 2.776 | 0.630 | 10.043 | 3.264 |
| **(I)** | PFSplat | **22.347** | **0.763** | **0.205** | **1.965** | **0.751** | 10.113 | 4.785 |
| **(II)** | - Depth Refinement | 21.963 | 0.759 | 0.208 | 2.240 | 1.134 | 9.701 | 3.044 |
| **(III)** | - Pose Refinement | 21.519 | 0.737 | 0.222 | 2.349 | 1.175 | 12.123 | 6.347 |
| **(IV)** | - Geometry Confidence | 21.239 | 0.737 | 0.223 | 2.303 | 0.922 | **9.179** | **3.533** |
| **(V)** | - Corres. Network | N/A | N/A | N/A | N/A | N/A | N/A | N/A |
| **(VI)** | - Mono. Depth Network | 16.132 | 0.511 | 0.405 | 6.990 | 5.329 | 21.328 | 14.432 |
| **(I-I)** | Full F.T. Depth Network | N/A | N/A | N/A | N/A | N/A | N/A | N/A |
| **(I-II)** | Scale/Shift Tuning Depth Network | N/A | N/A | N/A | N/A | N/A | N/A | N/A |
| **(I-III)** | - Tri. Consis. Loss | 18.832 | 0.6247 | 0.418 | 5.787 | 2.105 | 17.117 | 9.422 |
| **(I-IV)** | - Regularization Loss | 20.981 | 0.722 | 0.243 | 4.698 | 1.908 | 11.172 | 8.384 |
| **(I-V)** | (I-IV) - Tri. Consis. loss | N/A | N/A | N/A | N/A | N/A | N/A | N/A |

Table 4: **Component ablations on RealEstate10K.**

performance. Similar observations are made in (**I-I**), (**I-II**), and (**I-V**), where we identify that directly tuning the depth network or training only with photometric losses leads to failure in the training process. The former issue may arise from overfitting, a common problem when directly manipulating foundation models. With only the photometric loss, we observe that after certain iterations, as the baseline becomes wider, the training loss quickly diverges.

## 4.5 ANALYSIS AND MORE RESULTS

**Comparison to DUSt3R extensions.** In this study, we compare our results with those of InstantSplat (Fan et al., 2024) and Splatt3R (Smart et al., 2024). Although these are preprints, we include this comparison for completeness, as their tasks and methods are closely related to ours. The results are summarized in Tab. 5a. We find that InstantSplat achieves superior performance in novel view synthesis compared to our method; however, when we adopt a similar test-time optimization (TTO) strategy, our approach requires significantly less optimization time to achieve comparable or better results. Additionally, our method infers substantially faster without TTO, demonstrating its high practicality for real-time applications. Splatt3R, a concurrent work built on a similar concept to ours, requires ground-truth depth and pose for training, which are not available in the RealEstate10K dataset. Due to scale discrepancies, their performance is significantly lower, and without a mecha-

| Method | PSNR | SSIM | LPIPS | Rot. (°↓) Avg. | Med. | Trans. (°↓) Avg. | Med. | Time (s) |
|---|---|---|---|---|---|---|---|---|
| InstantSplat | 23.079 | 0.777 | **0.182** | 2.693 | 0.882 | 11.866 | 3.094 | 53 |
| Splatt3R | 15.636 | 0.502 | 0.360 | **1.312** | **0.521** | **8.715** | **1.891** | 20 |
| Ours | 22.347 | 0.763 | 0.205 | 1.965 | 0.751 | 10.113 | 4.785 | **0.390** |
| Ours + TTO | **23.132** | **0.779** | 0.202 | 1.965 | 0.814 | 9.996 | 4.701 | 13 |

(a) **Performance and speed comparisons on RealEstate-10K against DUSt3R variants.**

| Method | 2 views 1 view | 3 view | 5 view | 6 views 1 view | 3 view | 5 view | 12 views 1 view | 3 view | 5 view |
|---|---|---|---|---|---|---|---|---|---|
| DBARF | 1.456 | 4.562 | 8.177 | 2.965 | 7.288 | 13.780 | **4.009** | 10.493 | 17.50 |
| FlowCAM | 4.010 | 7.020 | 10.13 | 9.564 | 23.718 | 34.000 | 14.34 | 23.44 | 48.55 |
| CoPoNeRF | 17.29 | 33.78 | 54.52 | N/A | N/A | N/A | N/A | N/A | N/A |
| Ours | **0.390** | **0.392** | **0.394** | **2.054** | **2.056** | **2.058** | 5.725 | **5.727** | **5.729** |

(b) **Speed comparisons between pose-free generalizable view synthesis models.** Times are measured in seconds.

| Method | 6 views PSNR | SSIM | LPIPS | ATE | 12 views PSNR | SSIM | LPIPS | ATE |
|---|---|---|---|---|---|---|---|---|
| DBARF | 23.91662 | 0.7837 | 0.2226 | 0.0101166 | 24.1798 | 0.7906 | 0.2186 | 0.0048777 |
| FlowCAM | 24.6660 | 0.8259 | 0.2332 | 0.0022202 | 25.2290 | 0.8406 | 0.2169 | 0.0012655 |
| Ours | **26.3055** | **0.8664** | **0.1230** | **0.0011624** | **27.0822** | **0.8895** | **0.1055** | **0.0005684** |

(c) **RealEstate10K 6, 12 input views.**

| Method | RealEstate10K → DL3DV PSNR | SSIM | LPIPS | Rot.(°↓) Avg. | Med. | Trans.(°↓) Avg. | Med. | DL3DV → RealEstate10K PSNR | SSIM | LPIPS | Rot.(°↓) Avg. | Med. | Trans.(°↓) Avg. | Med. |
|---|---|---|---|---|---|---|---|---|---|---|---|---|---|---|
| MVSPlat | 23.543 | 0.764 | 0.179 | - | - | - | - | 22.198 | 0.760 | 0.211 | - | - | - | - |
| CoPoNeRF | 16.138 | 0.427 | 0.483 | 8.778 | 2.791 | 24.036 | 18.432 | 17.160 | 0.547 | 0.465 | 7.506 | 4.108 | 27.158 | 19.681 |
| Ours | 20.542 | 0.647 | 0.267 | 0.0672 | 0.0263 | 9.5373 | 5.3940 | 21.086 | 0.708 | 0.234 | 0.052 | 0.027 | 13.133 | 8.095 |

(d) **Cross-dataset Evaluation.**

Table 5: **More Analysis and Results.**

nism to handle the scale differences, an issue not addressed in the original paper, the results are not directly comparable to ours.

**Inference speed comparisons.** We conduct a comprehensive inference speed comparison between our method and competing approaches using varying numbers of input images, specifically $N \in 2, 6, 12$. For each scenario, we evaluate the time required to render 1, 3, and 5 views. The results, summarized in Tab. 5b, show that our approach is generally faster than existing methods. However, for $N = 12$, our inference speed is slower than that of DBARF, as our method involves estimating camera poses via a robust solver for every pairwise combination. Despite this overhead, our approach gains a significant advantage as the number of rendered views increases, due to the efficient rendering capabilities of 3DGS once the scene has been fully reconstructed. Finally, in order to provide a detailed breakdown, we measured three values for our method: overall inference time, UniDepth processing time, and decoder time, isolating the contributions of each component to the total runtime. Given 2, 6 and 12 views and to render a single target view, it takes 0.251, 0.832 and 1.535 seconds for UniDepth inference, while it takes approximately consistent 0.00247 seconds for rendering.

**Extending to N-views.** In practical scenarios, more than two views ($N > 2$) are commonly used. Therefore, we demonstrate that our method can process multiple views and render $I_t$. We input $N$ views into our network to obtain $\hat{\mathcal{P}}_i$, $\hat{\mathcal{D}}_i$, and $(\mu_i, \sigma_i, \Sigma_i, c_i)$. Following a similar approach to Chen & Lee (2023), we select the top-$k$ nearby views using $\hat{\mathcal{P}}_i$ and render $\hat{I}_t$ to compare with the ground truth target view image. For this evaluation, we compare our method with those of Chen & Lee (2023) and Smith et al. (2023), since the method by Hong et al. (2024) can only take two input views. We also report the Absolute Trajectory Error (ATE). The results are summarized in Tab. 5c. From these results, we find that our method achieves significantly better performance than the others, highlighting our capability to extend to multiple $N$ views.

**Cross-Dataset Evaluation.** To demonstrate the generalization capability of our method, we conduct a cross-dataset evaluation and compare against Hong et al. (2024). Specifically, we evaluate performance on RealEstate10K and DL3DV, using each dataset for training in a cross-dataset setting. The results, summarized in Tab. 5d, show that our method achieves a PSNR of over 20 dB for both datasets, significantly outperforming Hong et al. (2024). This indicates that, even under out-of-distribution conditions, our method produces high-quality renderings, highlighting its robustness and effectiveness in zero-shot capability.

## 5 CONCLUSION

In this paper, we have introduced learning-based framework that tackles pose-free novel view synthesis with 3DGS, enabling efficient, fast and photorealistic view synthesis from unposed images. The proposed framework, PFSplat, is built on the base model comprising of foundation models to overcome inherent limitations of 3DGS. While the devised base model already surpasses the existing methods, we have also devised modules to address the limitations of the base model, enhancing the overall performance. This method is capable of training and inference solely from unposed images, even in scenarios where only a handful of images with minimal overlaps are given. We have shown that our approach surpasses all existing methods on real-world large-scale datasets, establishing new state-of-the-art performance.

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

# A  APPENDIX

## A.1  TRAINING DETAILS

We train MVSplat (Chen et al., 2024) and CoPoNeRF (Hong et al., 2024) using our data loaders, similarly increasing the distance between context views during training, as explained in Sec 4.1. Specifically, we train MVSplat for 200,000 iterations using a batch size of 8 on a single A6000 GPU. All the hyperparameters are set to the authors' default setting. For CoPoNeRF, we train it for 50,000 iterations using 8 A6000 GPUs with effective batch size of 64, following the authors' original implementations and hyperparameters. Finally, for InstantSplat (Fan et al., 2024), we train and evaluate on a single A6000 GPU with a batch size of 1 by following the official code[1], and the hyperparameters were set according to the default settings provided by the authors.

## A.2  MORE QUALITATIVE RESULTS

Fig. 3, Fig. 4 and Fig. 6 present more novel view rendering results of different methods. On these datasets, our method yields outcomes that are sharper and more geometrically accurate.

## A.3  LIMITATIONS AND FUTURE WORK

As our model currently lacks a mechanism to handle dynamic scenes, it is unable to accurately capture scene dynamics or perform view extrapolation. Additionally, our model's performance is contingent on the quality of the coarse alignments, which rely on the accuracy of the depth and correspondence models. In cases where either of these models fails, our approach may not function optimally. However, because our refinement modules are lightweight, simple, and model-agnostic, incorporating more advanced methods for coarse alignment could further enhance performance.

For future work, we plan to train our model on diverse large-scale datasets. Since our approach relies exclusively on supervision signals from RGB images, it is straightforward to scale up the training data. We also aim to extend our method to handle 4D objects, ultimately enabling the modeling of 4D scenes, which would be beneficial for applications such as egocentric vision and robotics.

---

[1]https://github.com/NVlabs/InstantSplat

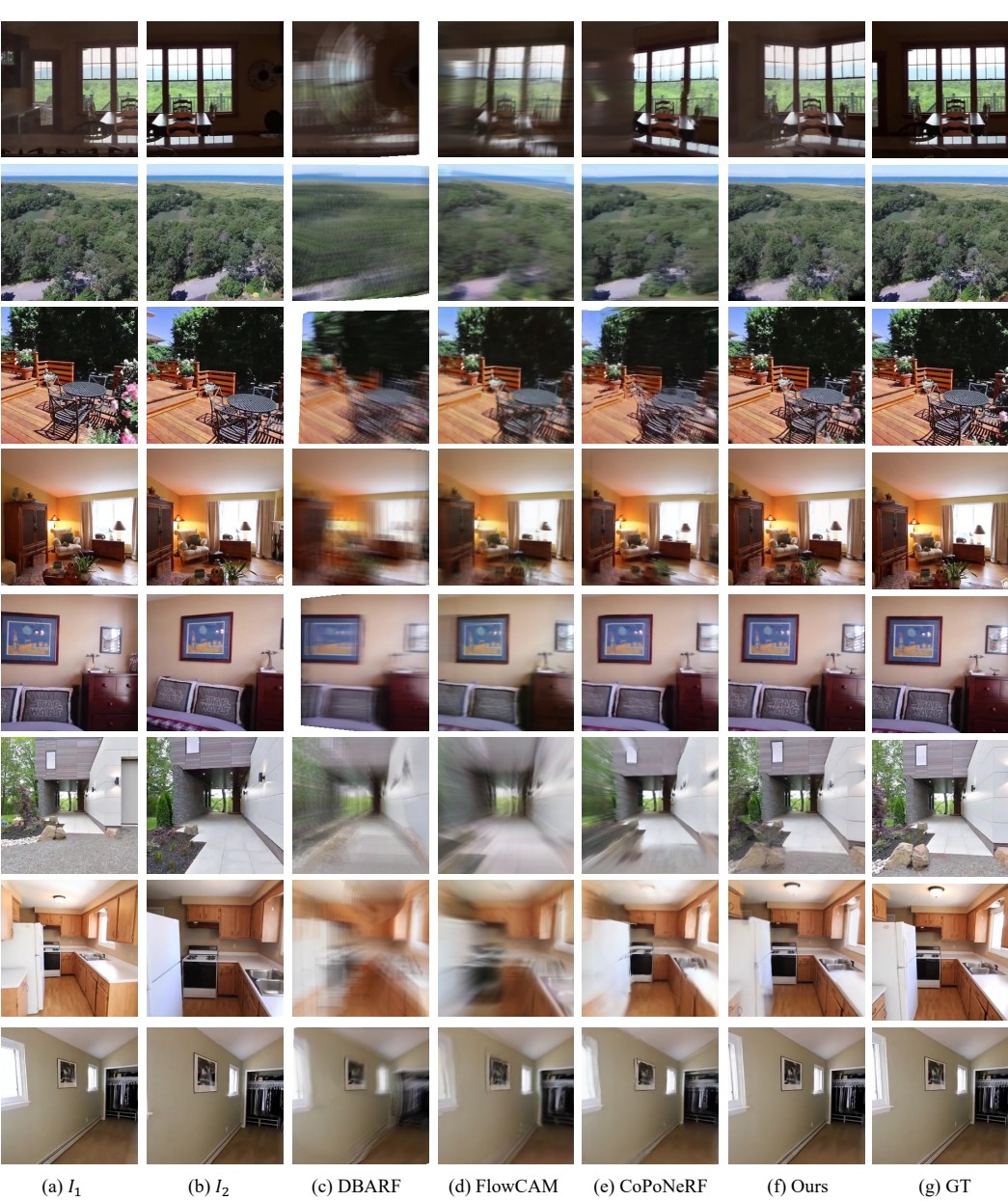

(a) $I_1$  (b) $I_2$  (c) DBARF  (d) FlowCAM  (e) CoPoNeRF  (f) Ours  (g) GT

Figure 3: **Qualitative results on RealEstate-10K dataset.** Given two context views (a) and (b), we compare novel view rendering results.

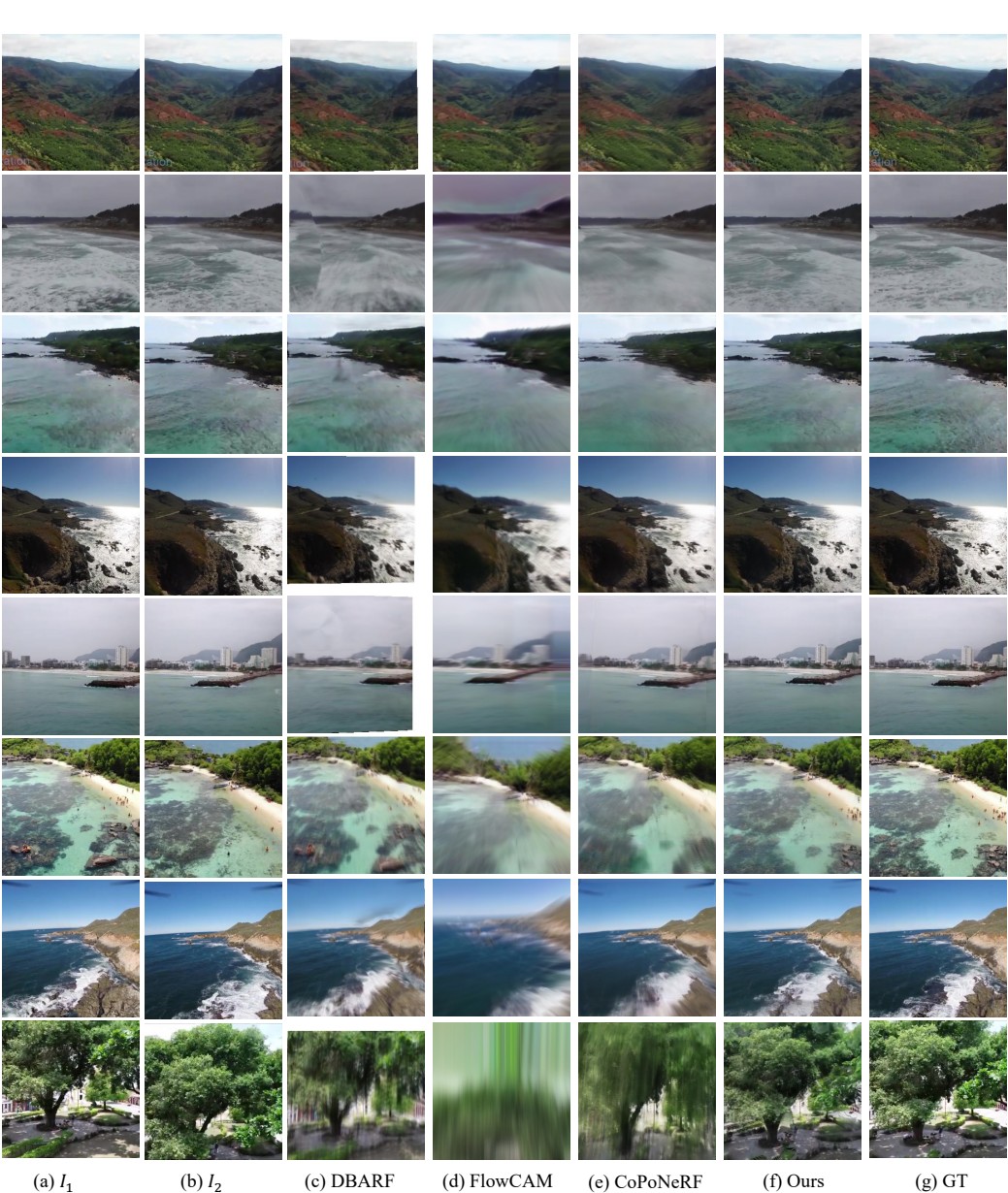

| (a) $I_1$ | (b) $I_2$ | (c) DBARF | (d) FlowCAM | (e) CoPoNeRF | (f) Ours | (g) GT |

Figure 4: **Qualitative results on ACID dataset.** Given two context views (a) and (b), we compare novel view rendering results.

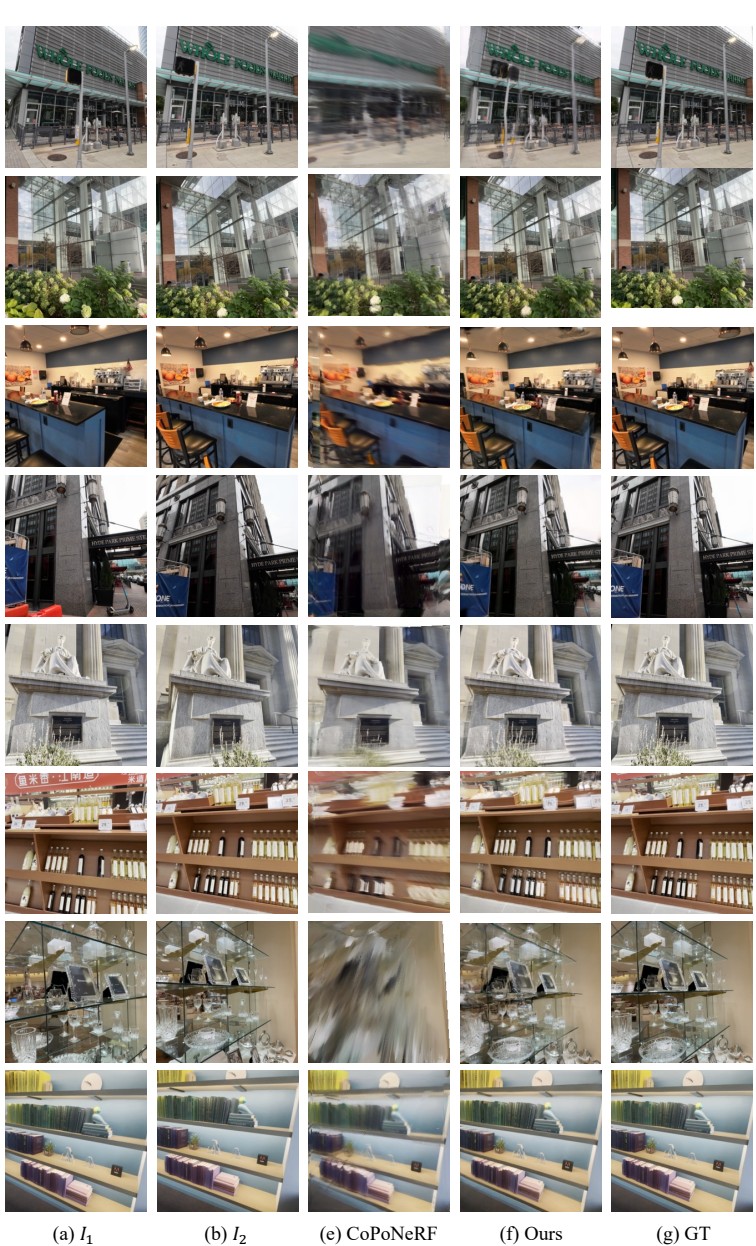

(a) $I_1$      (b) $I_2$      (e) CoPoNeRF      (f) Ours      (g) GT

Figure 5: **Qualitative results on DL3DV dataset.** Given two context views (a) and (b), we compare novel view rendering results.

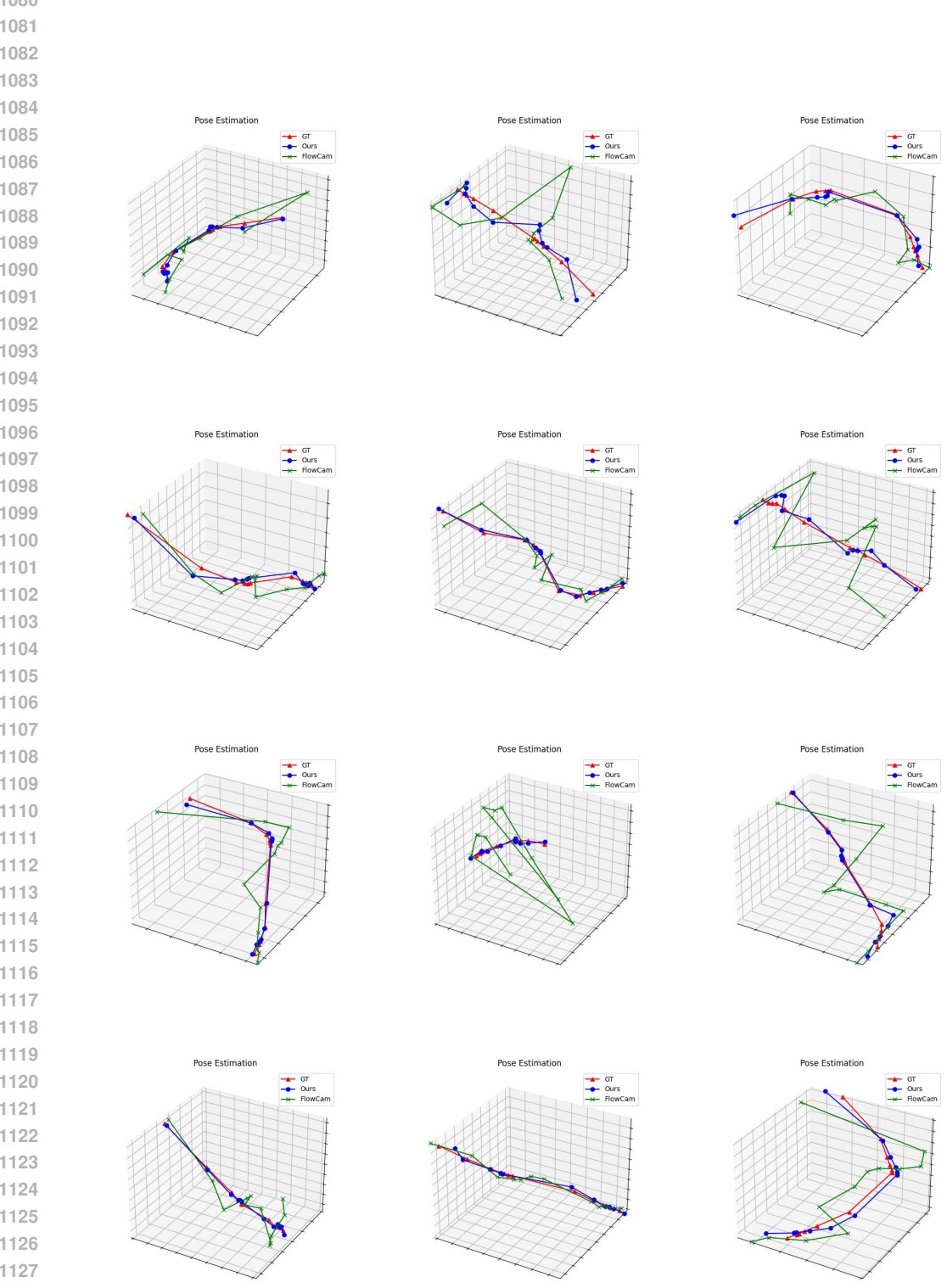

Figure 6: **Pose estimation qualitative results.**

