# OpenReview forum: "PF3plat: Pose-Free Feed-Forward 3D Gaussian Splatting"
_ICLR.cc/2025/Conference — Submitted to ICLR 2025_

### Official Review · Reviewer_HHva · 2024-10-23

**Soundness:** 2
**Presentation:** 3
**Contribution:** 3
**Rating:** 5
**Confidence:** 4

**Summary:**

Given G.T. camera intrinsics, the authors first leverage the existing work to obtain the coarse camera poses and depth. Then to refine the estimation, the authors design a module to learn the depth offset estimation with the help of an existing depth estimation network. Furthermore, the camera pose refinement is conducted in another module. The idea of feedforward pose estimation is interesting, but there is still a gap between the performance of the proposed method and some per-scene optimization methods. Since I did not see the authors report any inference time result and I believe some static scene pose-free per-scene optimization methods (CF-3DGS, ..) are very fast and accurate, I expect the authors to provide more comparisons. There are still some questions and limitations raised below. I will consider improving the grade upon the feedback from the authors.

**Strengths:**

1. The authors proposed a new pose-free feed-forward method in camera pose estimation.
2. The authors conducted enough ablation studies to present the contribution of each component of their method.

**Weaknesses:**

1. The biggest limitation of this work is the requirement of G.T. camera intrinsic.
2. The performance on RealEstate-10K seems to be SOTA, however, the performance on ACID is not. Does it mean such a method does not generalize well to the outdoor scenes?
3. So I expect more comparisons on DL3DV or more public datasets (like DAVIS[1], iPhone[2]) to prove the effectiveness of the proposed method.


[1] Pont-Tuset, Jordi, Federico Perazzi, Sergi Caelles, Pablo Arbeláez, Alex Sorkine-Hornung, and Luc Van Gool. "The 2017 davis challenge on video object segmentation." arXiv preprint arXiv:1704.00675 (2017).

[2] Gao, Hang, Ruilong Li, Shubham Tulsiani, Bryan Russell, and Angjoo Kanazawa. "Monocular dynamic view synthesis: A reality check." Advances in Neural Information Processing Systems 35 (2022): 33768-33780.

**Questions:**

1. Can the authors provide more insights on how you obtain the coarse camera parameters? I guess the authors directly implemented the existing work to obtain those things.
2. the authors trained different checkpoints on different datasets in the implementation details. Does it mean the 'feed-forward' claimed by the authors is actually dataset-specific? If so, I think it is a big limitation of the proposed method lacking the generalizability to different scenes.

---

> ### Author Response · Authors · 2024-11-21
> **Response to Reviewer HHva (1/3)**
>
> > The idea of feedforward pose estimation is interesting, but there is still a gap between the performance of the proposed method and some per-scene optimization methods. Since I did not see the authors report any inference time result and I believe some static scene pose-free per-scene optimization methods (CF-3DGS, ..) are very fast and accurate, I expect the authors to provide more comparisons.
>
> We kindly refer the reviewer to our table 5. (b), where we provide extensive speed comparison by varying both the number of input views and the number of rendering views. From the results, ours is the fastest.
>
> While we acknowledge the impressive performance of CF-3DGS, our method demonstrates notable advantages, particularly with wide-baseline images. Unlike scene-specific optimization methods, our approach performs inference in a single feed-forward pass, resulting in significantly faster processing speeds. This advantage is further highlighted in our comparison to InstantSplat in Table 5(a).
>
> To the best of our knowledge, scene-specific optimization literature has yet to comprehensively address the combination of wide-baseline, unposed, and sparse input image setups. Our method is the first to tackle all these challenges while achieving efficient, single-feed-forward inference with significantly faster speeds, making it a robust and practical solution for novel view synthesis.
>
> However, following reviewer's kind suggestion, we additionally compare with scene-specific optimization approaches like CF-3DGS, which can be found in the table below:
>
> | Method       | PSNR   | LPIPS | SSIM  | Rot (Avg) | Rot (Med) | Trans (Avg) | Trans (Med) | Time (s) |
> |--------------|--------|-------|-------|-----------|-----------|-------------|-------------|----------|
> | CF-3DGS      | 14.024 | 0.455 | 0.450 | 13.278    | 8.486     | 106.397     | 106.337     | 25       |
> | InstantSplat | 23.079 | 0.182 | 0.777 | 2.693     | 0.882     | 11.866      | 3.094       | 53       |
> | Ours         | 22.347 | 0.205 | 0.763 | 1.965     | 0.751     | 10.113      | 4.785       | 0.389    |
> | Ours + TTO   | 23.132 | 0.202 | 0.779 | 1.965     | 0.814     | 9.996       | 4.701       | 13       |
>
> From the results, our method consistently outperforms CF-3DGS in both image quality and pose accuracy. Our experiments reveal that CF-3DGS suffers from severe overfitting issues, likely due to its inability to effectively handle wide-baseline input image pairs—a scenario not addressed in its design. In contrast, our method demonstrates superior performance and efficiency, as evidenced in Tables 1, 2, 3, 5(a), and the additional results provided above.
>
> Furthermore, when test-time optimization is applied, our performance improves even further while requiring significantly less time, thanks to the geometry priors already learned during training. These results underscore the effectiveness and practicality of our proposed approach for wide-baseline, pose-free novel view synthesis.
>
> > The biggest limitation of this work is the requirement of G.T. camera intrinsic.
>
> While we acknowledge that the reliance on camera intrinsics could be viewed as a limitation, we would like to emphasize that this requirement is standard across all pose-free networks, including CF-3DGS. Nevertheless, we also acknowledge that modern smartphones and cameras typically provide intrinsic parameters, making them easily accessible at inference time. While there is active research [1,2] aimed at estimating camera calibration parameters, such advancements fall beyond the scope of this work. Nonetheless, we agree that integrating an intrinsic estimation process directly into our framework would enhance its practicality, and we view this as a promising direction for future research.
>
> [1] GeoCalib: Learning Single-image Calibration with Geometric Optimization, ECCV'24
>
> [2]  Perspective Fields for Single Image Camera Calibration, CVPR'23

---

> > ### Comment · Reviewer_HHva · 2024-11-25
> > **Response the authors' rebuttal**
> >
> > Thanks for the authors' rebuttal. I appreciate that it addresses some of my concerns, but some of my concerns are still valid.
> > 1) There are still many wonderful existing accepted papers with similar objectives to the authors' such as Vggsfm[1] and Dust3R[2]... Both of them do not require any camera intrinsic priors.
> > 2) I believe CoPoNeRF is not the current SOTA before the authors's submission. CF-3DGS, VGGSFM, DUST3R,... Also, there should be more experiments on more datasets. Please refer to the experimental datasets used in the works mentioned above. I believe 'SOTA on only one public dataset' is not enough.
> > 3) I do not think the authors include **the time to obtain the coarse camera parameters** in Table 5 (a). eg. '0.390s'.
> > 4) The detailed steps of obtaining the coarse camera parameters should be presented.
> > 5) I think Dust3R does not require to be trained on any specific dataset and can show great generalization to different datasets.
> >
> > I decide to keep my original points for now.
> >
> > [1] Wang, Jianyuan, et al. "VGGSfM: Visual Geometry Grounded Deep Structure From Motion." Proceedings of the IEEE/CVF Conference on Computer Vision and Pattern Recognition. 2024.
> >
> > [2] Wang, Shuzhe, et al. "Dust3r: Geometric 3d vision made easy." Proceedings of the IEEE/CVF Conference on Computer Vision and Pattern Recognition. 2024.

---

> ### Author Response · Authors · 2024-11-21
> **Response to Reviewer HHva (2/3)**
>
> > The performance on RealEstate-10K seems to be SOTA, however, the performance on ACID is not. Does it mean such a method does not generalize well to the outdoor scenes?
>
>
>
> We would like to clarify that our relatively lower performance on the ACID dataset can be attributed to the following factors: First, ACID predominantly consists of coastal imagery, characterized by much larger scene scales compared to conventional outdoor scene datasets such as driving scenes or cityscapes. Since our method relies on metric depth predictions from UniDepth to estimate camera poses, it may encounter difficulties in handling such large-scale scenes, which were not part of UniDepth’s training data. Effectively, for ACID, our method operates in a zero-shot depth estimation setting, which could explain the observed results.
>
> Additionally, ACID contains numerous dynamic and transient objects or regions, which may further impact the performance of our pose and depth estimation. Our current framework does not explicitly account for such dynamic elements, as this lies beyond the scope of this work.
>
> Nevertheless, as shown in the NVS performance results in Table 1, our method still outperforms other approaches, including CoPoNeRF, achieving state-of-the-art. This highlights the robustness of our method to noise in pose estimates, thanks to our proposed geometry-aware confidence estimation.
>
>
> > So I expect more comparisons on DL3DV or more public datasets (like DAVIS[1], iPhone[2]) to prove the effectiveness of the proposed method.
>
>
> We kindly refer the reviewer to our Table 3, where we already report our results on DL3DV and outperforms the previous state-of-the-art, CoPoNeRF, by significant margin. Furthermore, RealEstate10K and DL3DV, which comprise of approximately 28,000 and 12,000 scenes, offer a significantly larger variety of scenes and frames compared to most other datasets, such as DTU (80 scenes), Iphone (14 scenes) and DAVIS (50 scenes).
>
>
>
> Regarding the additional datasets, such as DAVIS or iPhone, these are dynamic scene datasets and do not align with the objectives of our work. Our method may struggle with these datasets, as it is specifically designed for static scene reconstruction and view synthesis, not for 4D scene reconstruction or dynamic view synthesis.
>
> We would also like to highlight that feed-forward 4D scene reconstruction remains largely unexplored in the community due to the highly challenging conditions it entails. Addressing moving objects would require an additional module to explicitly model the deformation of 3D Gaussians, a feature that is not yet implemented in this work. This is an active area of research, with some works focusing on explicitly estimating deformation fields for each 3D Gaussian [1, 2, 3] or removing transient objects [4, 5, 6]. However, modeling 4D scenes in a single-feed-forward setup remains an unexplored challenge. We appreciate the reviewer’s suggestion, as this represents a natural and promising direction for future development with potential practical applications in fields such as robotics and egocentric vision.
>
>
>
> [1] 4D Gaussian Splatting for Real-Time Dynamic Scene Rendering, CVPR'24
> [2] Real-time Photorealistic Dynamic Scene Representation and Rendering with 4D Gaussian Splatting, ICLR'24
> [3] Gaussian-Flow: 4D Reconstruction with Dynamic 3D Gaussian Particle, CVPR'24
> [4] WildGaussians: 3D Gaussian Splatting in the Wild, NeurIPS'24
> [5] Gaussian in the Wild: 3D Gaussian Splatting for Unconstrained Image Collections, ECCV'24
> [6] EgoLifter: Open-world 3D Segmentation for Egocentric Perception, ECCV'24
>
>
> > Question: Can the authors provide more insights on how you obtain the coarse camera parameters? I guess the authors directly implemented the existing work to obtain those things.
>
> We follow conventional pose estimation techniques to obtain coarse camera parameters. Given depth and correspondences, we can use methods such as PnP, Procrustes analysis, or the 8-point algorithm (in cases where depth is not used). In this paper, we specifically utilize the PnP algorithm with RANSAC for pose estimation.

---

> ### Author Response · Authors · 2024-11-21
> **Response to Reviewer HHva (3/3)**
>
> > the authors trained different checkpoints on different datasets in the implementation details. Does it mean the 'feed-forward' claimed by the authors is actually dataset-specific? If so, I think it is a big limitation of the proposed method lacking the generalizability to different scenes.
>
>
> We would first like to clarify that this scheme is a common practice in feed-forward NVS approaches. Methods are typically trained and evaluated on the same dataset while also being tested for generalization through cross-dataset evaluation. Notably, CoPoNeRF was the first work to train and evaluate on large-scale real-world datasets, such as RealEstate-10K and ACID, for the task of pose-free NVS, following the protocol established by Cross-Attention-Renderer [1]. To ensure consistency with the evaluation protocol and existing literature, we followed the same procedure in our work.
>
> However, in our work, we also provide quantitative results for cross-dataset evaluation, showcasing the generalization power of each method. We kindly refer the reviewer to Table 5(d), where we demonstrate that our proposed method significantly outperforms the previous state-of-the-art, CoPoNeRF, in cross-dataset scenarios. Furthermore, RealEstate10K and DL3DV, which we use in this work, comprise of approximately 28,000 and 12,000 scenes, offering a significantly larger variety of scenes and frames compared to most other datasets, such as DTU (80 scenes), Iphone (14 scenes) and DAVIS (50 scenes).  This demonstrates that our choice to evaluate on these large-scale real-world datasets show the effectiveness and robustness of our method across diverse real-world scenarios.
>
> [1] Learning to Render Novel Views from Wide-Baseline Stereo Pairs, CVPR'23

---

> ### Author Response · Authors · 2024-11-25
>
> Dear Reviewer HHva,
>
> As the author-reviewer discussion period is coming to an end, we wanted to kindly remind you of our responses to your comments. We greatly value your feedback and are eager to address any additional questions or concerns you might have.
>
> Please let us know if there's any further information we can provide to facilitate the discussion process.
>
> We are highly appreciated for  your time and consideration.

---

> ### Author Response · Authors · 2024-11-26
> **Response to the reviewer HHva (1/2)**
>
> We are delighted to find that our response addressed some of the reviewer's concerns, and we would like to further clarify the remaining ones. Before addressing these points, we wish to emphasize that our primary focus lies on **Generalized Novel View Synthesis/Rendering from unposed images, as clearly stated in the title, introduction, and contributions (L88)**. Pose Estimation, as well as other tasks such as depth estimation, correspondence estimation,  are the intermediate tasks required for our final objective, novel view rendering from unposed images.
>
> Unlike SfM-related works, such as Dust3R, VGGSfM, SP+SG, or other correspondence-based 3D reconstruction methods, our approach differs in terms of data, objectives, and training and evaluation setups. Generalized NVS approaches, pioneered by PixelNeRF and followed by works like MVSNeRF and IBRNet, introduced effective strategies for photorealistic rendering applicable to applications such as VR/AR.
>
> Building upon these foundations, pose-free scene-specific optimization NVS methods (e.g., BARF, Nope-NeRF) emerged, and more recently, methods like DBARF, FlowCAM, and CoPoNeRF have advanced the field of generalized pose-free NVS. Our work falls within this topic, with a key distinction from existing SfM methods being our focus on view synthesis for rendering photorealistic novel views, which inherently targets different objectives than SfM tasks.
>
> An easy example would be prior to NeRF and 3DGS, which focus on accurate camera pose estimation or SfM initialization, areas where the reviewer’s mentioned works excel. Our approach, however, directly and additionally tackles the downstream task of novel view rendering without requiring these preconditions beforehand, thereby relaxing many of the common assumptions (L14) and addressing a distinct challenge.
>
> > There are still many wonderful existing accepted papers with similar objectives to the authors' such as Vggsfm[1] and Dust3R[2]... Both of them do not require any camera intrinsic priors.
>
> In this regard, we wish to clarify that VGGSfM and Dust3R are **methods exclusively addressing 3D reconstruction**, and additional methods, such as NeRF or 3DGS, are required to train a radiance field for novel view synthesis. It is important to note that this 1) **is not a feature of VGGSfM or Dust3R**, 2) **requires a long optimization time for each scene**, and 3) **depends on ground-truth camera pose, tracking, intrinsics and depth for training**.
>
> While we acknowledge that our method also relies on ground-truth intrinsics, this requirement is shared by all other NVS approaches, including PixelSplat, MVSplat, DBARF, FlowCAM, and CoPoNeRF. As previously discussed, we consider the exploration of intrinsic camera parameter estimation as a promising direction for future work.
>
> > I believe CoPoNeRF is not the current SOTA before the authors's submission. CF-3DGS, VGGSFM, DUST3R,... Also, there should be more experiments on more datasets. Please refer to the experimental datasets used in the works mentioned above. I believe 'SOTA on only one public dataset' is not enough.
>
> We highlight that for novel view synthesis, it is crucial to distinguish that CF-3DGS is an example of a scene-specific optimization approach for pose-free NVS, and it is not considered SOTA in terms of both performance and speed, as previously addressed in our response. Similarly, VGGSfM and Dust3R tackle different tasks (which reviewer 943i acknowledges this after our response and other reviewers all acknowledge that these are different tasks' methods), making CoPoNeRF the SOTA for generalized pose-free NVS.
>
> Furthermore, as already stated, we followed the **same** experimental datasets used by CoPoNeRF and Cross-Attention Renderer (noting that PixelSplat and MVSplat also used the same datasets). We have provided sufficient experimental evaluations to validate our approach and additionally included DL3DV dataset evaluations, which were not used by previous NVS methods.
>
> We also wish to clarify that our approach achieves SOTA across multiple public datasets and **is not limited to a single benchmark**. If evaluated on benchmarks commonly used by SfM-like methods, such as MegaDepth or HLoc, our method would require substantial modifications to adapt to their specific focus on 3D reconstruction tasks rather than novel view synthesis. However, these benchmarks are not representative of the challenges posed by the pose-free generalized NVS task, which requires simultaneous pose estimation, depth estimation, and photorealistic rendering. Our results on RealEstate-10K, ACID, and DL3DV, **which are large-scale and real-world datasets containing much more various scenes** than DAVIS or IPhone,  highlight our method's effectiveness in addressing these challenges on datasets explicitly designed for the NVS task.

---

> ### Author Response · Authors · 2024-11-26
> **Response to the reviewer HHva (2/2)**
>
> > I do not think the authors include the time to obtain the coarse camera parameters in Table 5 (a). eg. '0.390s'.
>
> We wish to seek clarification from the reviewer regarding this point. The speed comparison in Table 5(a) represents the overall time consumption, which includes UniDepth, LightGlue, the robust solver with RANSAC, and the differentiable rasterizer for view rendering, as mentioned in L511. We would like to clarify that the reported 0.390s **does include the time to obtain coarse camera parameters.**
>
> If the reviewer is specifically referring to the time for the robust solver with RANSAC alone (L506), we note that this step takes less than 0.1s, as UniDepth itself requires 0.251s (L511). We kindly ask the reviewer to confirm or clarify his/her interpretation so we can address this point further if needed.
>
> > The detailed steps of obtaining the coarse camera parameters should be presented.
>
> We simply follow the widely known procedure for camera pose estimation. Given the estimated depth map, a set of 3D points is generated in the camera coordinate system, and corresponding 2D points are extracted using a feature-matching algorithm, such as LightGlue. Using the known camera intrinsics, these 3D-2D correspondences are passed to a PnP solver, which computes an initial estimate of the camera pose, including rotation and translation.
>
> To ensure robustness, the PnP estimation is integrated with RANSAC, where multiple subsets of the correspondences are sampled to generate pose hypotheses. Each hypothesis is evaluated by calculating the reprojection error, and inliers are identified as correspondences with reprojection errors below a predefined threshold. The pose hypothesis with the largest inlier set is selected as the best estimate, and a final optimization step using the inlier correspondences refines the camera pose.
>
> This process yields a robust and accurate coarse camera pose, mitigating the impact of outliers in the correspondence set.
>
> We will include this in the supplementary material.
>
> > I think Dust3R does not require to be trained on any specific dataset and can show great generalization to different datasets.
>
> Our method demonstrates **superior performance compared to Dust3R**, and when combined with RoMa, it surpasses the advanced version, Mast3R. Notably, our approach achieves these results while being trained solely with RGB images, **without relying on ground-truth pose or depth specifically tailored for correspondence or pose estimation tasks**.
>
> The table below presents the comparison:
>
> | Method       | PSNR   | SSIM  | LPIPS | Rot (Avg) | Rot (Med) | Trans (Avg) | Trans (Med) |
> |--------------|--------|-------|-------|-----------|-----------|-------------|-------------|
> | Mast3R       | -      | -     | -     | 2.555     | 0.751     | 9.775       | 2.830       |
> | Ours         | 22.347 | 0.763 | 0.205 | 1.965     | 0.751     | 10.113      | 4.785       |
> | Ours (RoMa)  | 23.121 | 0.798 | 0.191 | 1.874     | 0.723     | 7.674       | 3.891       |
>
> These results highlight the strong generalization capability of our method in the NVS task. Furthermore, we anticipate that our coarse alignment approach can generalize effectively to other datasets as well. However, as our primary focus is on **novel view rendering**, we believe that our experiments in Table 5(d) clearly showcase the generalization power of our method in this domain.

---

> > ### Author Response · Authors · 2024-11-29
> >
> > Dear Reviewer HHva,
> >
> > We have provided additional explanations and experimental results to make our contributions and work’s focus more clear. As the author-reviewer discussion period is coming to an end, we would appreciate it if the reviewer could take a look at our responses. We greatly value your feedback and are eager to address any further concerns or questions.

---

### Official Review · Reviewer_RiDd · 2024-11-02

**Soundness:** 3
**Presentation:** 3
**Contribution:** 3
**Rating:** 6
**Confidence:** 3

**Summary:**

This paper focuses on the pose-free feed-forward novel view synthesis task. It leverages a pre-trained monocular depth estimation model and a visual correspondence model to generate coarse pose and depth estimates, enhancing the stability of the 3DGS training process. Subsequently, a lightweight refinement model is used to further improve the depth and pose estimations. Extensive experiments have been conducted to demonstrate the effectiveness of the proposed method.

**Strengths:**

S1 - This paper addresses a meaningful task with significant potential for real-world applications.

S2 - This paper leverages two robust pre-trained models to generate initial pose and shape estimates, which significantly enhance the model's performance.

S3 - The paper is well-written, and the experiments are logically sound.

**Weaknesses:**

W1 - The performance of this method appears to be highly dependent on the coarse pose and depth results provided by the pre-trained model.

W2 - The paper lacks qualitative results for the pose estimation, which would provide a clearer assessment of the model's performance in this area.

**Questions:**

Q1 - How would the results differ if alternative coarse prediction networks, such as Dust3r[1], Mast3r[2], or others, were used?

Q2 - Qualitative results for the pose estimation task.

[1]Wang S, Leroy V, Cabon Y, et al. Dust3r: Geometric 3d vision made easy[C]//Proceedings of the IEEE/CVF Conference on Computer Vision and Pattern Recognition. 2024: 20697-20709.
[2]Leroy V, Cabon Y, Revaud J. Grounding Image Matching in 3D with MASt3R[J]. arXiv preprint arXiv:2406.09756, 2024.

---

> ### Author Response · Authors · 2024-11-21
> **Response to Reviewer RiDd (1/1)**
>
> We are highly thankful for the reviewer's insightful comments. Our responses can be found below.
>
> > Question:  How would the results differ if alternative coarse prediction networks, such as Dust3r[1], Mast3r[2], or others, were used?
>
> While we acknowledge that incorporating predictions from networks like Mast3r or Dust3r could potentially improve performance, seamlessly integrating these methods into our framework, extending Dust3r or Mast3r to support
> N-view feed-forward inference without relying on global optimization (the inference time significantly increases, sacrificing the advantage of our approach), and addressing the associated implementation complexities may not be feasible within the rebuttal period.
>
> To highlight the flexibility of our framework, we instead provide results that incorporate alternative monocular depth estimators and correspondence estimators. This integration is straightforward due to our framework's adaptability to different pretrained models. The results are presented below and we believe below results offer insights of differing results based on alternative coarse prediction network:
>
> | Method          | PSNR   | SSIM  | LPIPS | Rot (Avg) | Rot (Med) | Trans (Avg) | Trans (Med) |
> |-----------------|--------|-------|-------|-----------|-----------|-------------|-------------|
> | Ours (DepthPro) | 22.127 | 0.744 | 0.218 | 2.115     | 0.772     | 11.484      | 5.882       |
> | Ours (RoMa)     | 23.121 | 0.798 | 0.191 | 1.874     | 0.723     | 7.674       | 3.891       |
> | Ours            | 22.347 | 0.763 | 0.205 | 1.965     | 0.751     | 10.113      | 4.785       |
>
>
>
> Interestingly, while we observe improvements when RoMa is incorporated, we note a slight performance degradation when using DepthPro. This degradation could be attributed to the selection of feature layers used or the inherent performance differences between DepthPro and UniDepth. We thank the reviewer for this insightful comment and will include these discussions in the supplementary material.
>
>
> > The performance of this method appears to be highly dependent on the coarse pose and depth results provided by the pre-trained model.
>
> We acknowledge our method may have dependency on its performance on coarse alignment. However, as shown in Table 4, the effectiveness of **our proposed lightweight refinement modules that enable apparent performance improvements**. We also show that with our proposed design, we show that ours outperforms existing methods by a significant margin.  We would also like to emphasize the flexibility of our approach, which allows us to integrate various models, offering the advantage of selecting the most suitable model for specific scenarios.
>
>
> > The paper lacks qualitative results for the pose estimation, which would provide a clearer assessment of the model's performance in this area
>
>
> Thank you for the suggestion. While it is certainly possible to visualize epipolar lines or compare the locations of predicted and ground-truth camera poses, as done in CoPoNeRF or FlowCAM, we wish to  highlight that in our case, where 3D Gaussian Splatting is used for scene representation, pose estimation directly determines the location of each pixel-wise 3D Gaussian, meaning the quality of the estimated poses is inherently tied to the final rendering results. This makes rendering outcomes the one of the most appropriate visualizations to evaluate the quality of both pose estimates and the radiance field.
>
> However, we agree that including pose estimation qualitative results can further strengthen our work, and following the reviewer's kind suggestion, we have updated our submission paper and included the pose estimation visualization. Please see Fig. 6 in the updated PDF.

---

> ### Author Response · Authors · 2024-11-25
>
> Dear Reviewer RiDd,
>
> As the author-reviewer discussion period is coming to an end, we wanted to kindly remind you of our responses to your comments. We greatly value your feedback and are eager to address any additional questions or concerns you might have.
>
> Please let us know if there's any further information we can provide to facilitate the discussion process.
>
> We are highly appreciated for  your time and consideration.

---

> > ### Comment · Reviewer_RiDd · 2024-11-25
> >
> > Thank you for your response. I have another question: will your method be more effective than the combination of readily available models designed for separate tasks (e.g., Mast3R and MVSplat)? This point has been addressed in CoPoNeRF, and I believe it is crucial to demonstrate the significance of this combined training strategy over handling tasks separately. It is possible to directly utilize the checkpoints from MVSplat and the pose estimation results from Mast3R without the need for additional training.

---

> > > ### Author Response · Authors · 2024-11-25
> > >
> > > Thank you for your response!
> > >
> > > We agree that CoPoNeRF effectively addressed the reviewer's point and emphasized its importance throughout the paper.
> > >
> > > To align with this, we included similar results for our coarse alignment + MVSplat in Table 4(0) and progressively added our proposed modules to demonstrate the improvements. If the reviewer considers it necessary, we will also include the suggested baseline (Mast3R + MVSplat).

---

> > > > ### Comment · Reviewer_RiDd · 2024-11-26
> > > >
> > > > Thank you for your response. I still believe that this baseline (Mast3R + MVSplat) is crucial for highlighting the significance of the combined training strategy. It would be better if you can show this result.

---

> ### Author Response · Authors · 2024-11-29
>
> We greatly appreciate the reviewer's suggestion. In response, we conducted evaluations on the standard DL3DV benchmark to avoid the excessive time consumption associated with Mast3R's inference time and the large number of scenes in the RealEstate10K test split, which would require several days for a full evaluation.
>
> First, we compared the results of different variants to highlight the significance of our training approach. For this experiment, we used pre-trained weights obtained from training on DL3DV and evaluated on the same dataset. The results are shown below :
>
> | Methods | PSNR $\uparrow$ | SSIM $\uparrow$  | LPIPS $\downarrow$ | Time (s) $\downarrow$ |
> | --- | --- | --- | --- | --- |
> | (0) Ours Pose + MVSplat | 18.811 | 0.578 | 0.342 | **0.264** |
> | Mast3R Pose + MVSplat | 19.416 | 0.599 | 0.318 | 18 |
> | Mast3R (pnp + ransac) Pose + MVSplat | 18.833 | 0.589 | 0.399 | 0.642 |
> | Ours | **20.730** | **0.659** | **0.225** | 0.390 |
>
> From the results, we observe that our method achieves better scores than the other variants. Notably, we included a variant that incorporates Mast3R's pose prediction without performing any optimization (since Mast3R's default setting performs per-scene optimization). We found that this variant is on par with our baseline (0) in performance, while the original Mast3R's optimization time takes approximately ~18 seconds for optimization, making the overall inference time take ~40 seconds, which significantly affects speed.
>
> Additionally, we conducted evaluations in a cross-dataset setting, where models pre-trained on RealEstate10K are evaluated on DL3DV. In real-world scenarios, systems are expected to perform inference on out-of-domain data, not just on the data they were trained on. Moreover, such target evaluation data typically lacks dense views suitable for Structure-from-Motion (SfM) or Simultaneous Localization and Mapping (SLAM), making it difficult to acquire camera poses and thus rendering training on this domain infeasible. By entirely disallowing access to ground-truth camera poses in the target domain, this setup better aligns with our goal of practical, pose-free generalized novel view synthesis. The results are shown below:
>
> | Methods | PSNR $\uparrow$ | SSIM $\uparrow$  | LPIPS $\downarrow$ | Time (s) $\downarrow$ |
> | --- | --- | --- | --- | --- |
> | (0) Ours Pose + MVSplat | 18.642 | 0.548 | 0.383 | **0.264** |
> | Mast3R Pose + MVSplat | 20.082 | 0.622 | 0.288 | 18 |
> | Mast3R (pnp + ransac) Pose + MVSplat | 19.198 | 0.579 | 0.392 | 0.642 |
> | Ours | **20.542** | **0.647** | **0.267** | 0.390 |
>
> From the table above, we also observe similar results. We again thank the reviewer for the question and we will include this discussion in the final version of our pdf.

---

### Official Review · Reviewer_943i · 2024-11-03

**Soundness:** 2
**Presentation:** 2
**Contribution:** 2
**Rating:** 5
**Confidence:** 5

**Summary:**

This paper introduces a method to tackle the challenging problem of novel view synthesis (NVS) from unposed images in a feed-forward manner. They identify the issue in previous pixel-aligned 3DGS methods, where the predicted 3D Gaussians from different views have the problem of misalignment, leading to noisy gradient flows and poor NVS results. They propose a method where they don’t need to rely on the poses obtained from off-the-shelf tools. Instead, they leverage pre-trained monocular depth estimation and visual correspondence models to obtain coarse alignment, with further refinement of the depth map and poses. Results show that among the pose free methods, they can perform decently better results for NVS tasks. However, for pose estimation, it is still worse than general methods like Mast3R.

**Strengths:**

1. The authors tackle the problem of 3D Gaussian splatting from unposed sparse images, which is an interesting and important topic
2. The authors apply the recent state-of-the-art depth estimation and pose estimation methods for coarse pose alignment, and further introduces pose and depth refinements, to some extend improving the final performance.
3. The paper is overall well-written and easy to follow in most parts.

**Weaknesses:**

**The method section is overall clear, but missing some details and discussions**

*Unclear descriptions in camera pose refinement in Sec 3.2.3*
1. It is quite unclear to me how exactly it is done. From your writing, it makes me feel that you first will get a newly computed camera poses $\hat{P_{ij}}$ similarly as done in the coarse step, but with the refined depth. This $\hat{P}_{ij}$ is already the refined poses, right? However, you further have another refinement step as shown in Eq. (2). What are the rationale before?
2. what is the T_pose network? What is the E_pos in eq. (2)?

*Cost volume*
In section 3.2.4, for the “cost volume construction and aggregation”, is that any different to the MVSplat paper? Can you justify the differences?

*2D-3D Consistency loss*
Line 291-292, you said that you improve the robustness of the model in regions with low texture or significant viewpoint changes. However, the correspondences from the feature matching methods like LightGlue do not provide many correspondences in those low-texture regions. I guess you cannot claim the robustness there?

*Unclear implementation details*
What is the frame distance from 45 to 75? Do you mean you sample one frame every 45/75 frames in the video sequence? You might want to make this point clearer. And why for DL3DV, you only sample every 5 or 10 frames, way smaller than ACID or RE10LK?
Also, you mention that you train for 40000 iterations on a single A6000 GPU, is it the same for all three datasets? If true, I think it might not make too much sense? As you mentioned in line 351-354, RE10K has ~21K videos from training, while you only use 2K scenes for the training of DL3DV?



**The experimental section is convincing in general, but lacks some important experiments / baselines and explanation.**

1. For pose estimation comparison, the sota method right now is RoMa [1], I think it is fair to ask to compare to it.
2. I wonder why your method is still lacking behind Mast3R on the pose estimation for both RE10K and ACID, even if Mast3R is is not even trained at all on those dataset, while your method was trained on those dataset separately.
3. Why for novel view synthesis in Table 1, you don’t show the comparison to the recent pose-required methods pixelSplat?
4. Based on the experiments you show, your pose estimation is worse than Mast3R in almost all cases, and the NVS results are also worse than MVSplat in all scenarios (even on DL3DV in Table 3, where MVSplat was not trained on). One reasonable baseline to me is, get camera poses from Mast3R, and then run MVSplat directly. I wonder how your method compares to such a simple baseline?
5. In your ablation study table 4, what is the point of adding V, I-I, I-II, I-V, if they are just all N.A.? That is really weird to me. You can just describe them in texts.


**Writing**
Sec 3.2.1: You use two paragraphs to motivate by mentioning the limitations of the previous methods. The real content for the coarse alignment is really just the third paragraph between line 186-191. The motivations part is actually unrelated to the coarse alignment but more on why your method is needed, so you should just put them in the introduction instead.

[1] Edstedt et al.: RoMa: Robust dense feature matching, CVPR 2024

**Questions:**

As I already discussed in the weakness section, it would be very imporatnt if you can justify those points in the experiments.

---

> ### Author Response · Authors · 2024-11-21
> **Response to Reviewer 943i (1/3)**
>
> We appreciate the reviewer's thorough comments that can certainly improve our paper once addressed. Our responses can be found below.
>
> > It is quite unclear to me how exactly it is done. From your writing, it makes me feel that you first will get a newly computed camera poses
>  similarly as done in the coarse step, but with the refined depth. This
>  is already the refined poses, right? However, you further have another refinement step as shown in Eq. (2). What are the rationale before?
>
>
> Your understanding is correct. We first compute a newly refined camera pose using the refined depth, which can indeed be considered a "refined pose." However, we further refine this pose using our proposed refinement module. The rationale behind this two-step process is as follows: while the refined pose derived from better depth estimation already improves performance, our refinement module directly targets enhancing the pose estimation further. Thus, the first stage focuses on refinement through improved depth estimation, while the second stage directly optimizes the camera pose itself.
>
> For clarity, we have included an additional figure in the paper (See Fig.2) to illustrate this process. Thank you for your question!
>
>
> > what is the Tpose network? What is the Epos in eq. (2)?
>
> Tpose refers to a transformer-based architecture, while Epos denotes the positional embedding used in the model. We apologize for the missing explanation in the initial submission and have included a newly added figure (See Fig. 2) to clarify these components. Thank you for pointing this out!
>
> > In section 3.2.4, for the “cost volume construction and aggregation”, is that any different to the MVSplat paper? Can you justify the differences?
>
> For the multi-view cost volume, we follow the conventional plane-sweeping approach, as cited in L246. The key difference from MVSplat, which constructs a conventional plane-sweeping cost volume for depth estimation, lies in the construction and use of an additional guidance cost volume, where each spatial location is represented by a one-hot vector indicating the depth candidate closest to the monocular depth estimate. In our method, we **aggregate the multi-view stereo cost volume with the guidance cost volume** to obtain a confidence score. This process leverages the estimated depth and camera pose to compute the confidence score, which is a distinct departure from MVSplat’s approach of constructing a multi-view plane-sweeping cost volume with ground-truth camera poses to estimate depth.
>
> We kindly direct the reviewer to our newly added figure (See Fig. 2) and the explanation provided in L254–L264 for further details.
>
> > Line 291-292, you said that you improve the robustness of the model in regions with low texture or significant viewpoint changes. However, the correspondences from the feature matching methods like LightGlue do not provide many correspondences in those low-texture regions. I guess you cannot claim the robustness there?
>
> We acknowledge that in scenarios where LightGlue struggles to establish correspondences, our method may not perform as robustly as expected. However, as demonstrated in the LightGlue paper, it generally exhibits strong robustness. More importantly, our approach is flexible and can seamlessly incorporate alternative correspondence-based methods, such as RoMa, to enhance performance.
>
> The key advantage of our proposed method is its adaptability, it is not restricted to specific components like UniDepth or LightGlue. If improved robustness is required, we can readily substitute LightGlue with a more robust correspondence network. To illustrate this, the table below shows how our method benefits from using a more robust correspondence estimator, RoMa:
>
> | Method       | PSNR   | SSIM  | LPIPS | Rot (Avg) | Rot (Med) | Trans (Avg) | Trans (Med) |
> |--------------|--------|-------|-------|-----------|-----------|-------------|-------------|
> | RoMa         | -      | -     | -     | 2.470     | 0.391     | 8.047       | 1.601       |
> | Ours         | 22.347 | 0.763 | 0.205 | 1.965     | 0.751     | 10.113      | 4.785       |
> | Ours (RoMa)  | 23.121 | 0.798 | 0.191 | 1.874     | 0.723     | 7.674       | 3.891       |
>
> From the results, we observe apparent performance improvements, highlighting the advantage of our approach. Note that the tendency for lower average errors but higher median errors compared to matching-based methods can be attributed to observations made in [1], where robust solvers tend to estimate precise poses. In contrast, learning-based approaches like ours produce more robust pose estimates, leading to greater consistency across diverse scenarios.
>
>
> [1] FAR: Flexible, Accurate and Robust 6DoF Relative Camera Pose Estimation, CVPR'24

---

> ### Author Response · Authors · 2024-11-21
> **Response to Reviewer 943i (2/3)**
>
> > What is the frame distance from 45 to 75? Do you mean you sample one frame every 45/75 frames in the video sequence? You might want to make this point clearer. And why for DL3DV, you only sample every 5 or 10 frames, way smaller than ACID or RE10LK? Also, you mention that you train for 40000 iterations on a single A6000 GPU, is it the same for all three datasets? If true, I think it might not make too much sense? As you mentioned in line 351-354, RE10K has ~21K videos from training, while you only use 2K scenes for the training of DL3DV?
>
> During training, we initially set the frame distance between
> I1 and I2
>   to 45 frames and gradually increase it to 75 frames. The target view is sampled between
> I1 and I2. The variation in frame distance arises from differences in the speed of viewpoint changes across datasets. Empirically, we found that for DL3DV, a frame distance of 5 or 10 frames provides similar viewpoint changes to those in other datasets.
>
> Regarding the fixed training iterations, we seek clarification from the reviewer on whether the suggestion is to train each dataset for a proportional number of iterations based on the total number of scenes (e.g., ensuring
> 21K/N for RE10K,
> 11K/N for ACID, and
> 2K/N for DL3DV produce equivalent numbers). While this approach could balance scene counts, it is impractical to implement differing iterations for each dataset. It is standard practice to train models with a fixed number of iterations across datasets, as has been done for baselines such as PixelSplat, MVSplat, and CoPoNeRF, which also trained for the same number of iterations on both RE10K and ACID.
>
>
>
>
>
> > For pose estimation comparison, the sota method right now is RoMa [1], I think it is fair to ask to compare to it.
>
> We wish to emphasize, as stated in L88–94, that **our primary contribution lies in feed-forward novel view synthesis with 3DGS, addressing challenges specific to the pose-free feed-forward task, and proposing a refinement module to enhance performance**. While pose estimation is undoubtedly important, our goal is not to achieve state-of-the-art (SOTA) performance across all related tasks but to tackle the unique challenges of pose-free feed-forward synthesis with 3DGS and present innovative solutions within this scope.
>
> As noted in L392–394, the gray entries in the tables are included for reference only, as they are not trained on the same dataset due to the absence of ground-truth data. Unlike methods requiring ground-truth correspondence or depth, our approach does not rely on such data, which differentiates it in terms of training data, labels, and objectives. Methods like RoMa require ground-truth depth or matches, while our framework requires ground-truth novel views, making direct comparisons misleading. However, we could seamlessly incorporate ground-truth depth or matches into our unified framework to enhance performance, should such data be available.
>
> Finally, RoMa, as a 2D correspondence network, faces inherent challenges with unknown scale when estimating camera poses using the 8-point algorithm, particularly in scenarios without depth information. This limitation makes it less suitable for our specific task of pose-free novel view synthesis when the depth information is not accounted for. While the reviewer's suggestion to include more references is reasonable, we believe these comparisons should serve only as references.
>
> Nevertheless, we report RoMa's results on our evaluation split, which can be found below:
>
> | Method       | PSNR   | SSIM  | LPIPS | Rot (Avg) | Rot (Med) | Trans (Avg) | Trans (Med) |
> |--------------|--------|-------|-------|-----------|-----------|-------------|-------------|
> | RoMa         | -      | -     | -     | 2.470     | 0.391     | 8.047       | 1.601       |
> | Ours         | 22.347 | 0.763 | 0.205 | 1.965     | 0.751     | 10.113      | 4.785       |
> | Ours (RoMa)  | 23.121 | 0.798 | 0.191 | 1.874     | 0.723     | 7.674       | 3.891       |
>
>
> > Why for novel view synthesis in Table 1, you don’t show the comparison to the recent pose-required methods pixelSplat?
>
> While we can include PixelSplat for comparison, we believe that including MVSplat already provides a sufficient reference to indicate the difficulty of the test set. It is important to note that PixelSplat is a pose-required method, which differs from our task. Our method addresses a pose-free task, making direct comparisons to PixelSplat less relevant. Nevertheless, as the reviewer kindly suggests, we include PixelSplat's results below:
>
> | Method     | PSNR   | SSIM  | LPIPS |
> |------------|--------|-------|-------|
> | PixelSplat | 24.788 | 0.819 | 0.176 |
> | Ours       | 22.347 | 0.763 | 0.205 |

---

> ### Author Response · Authors · 2024-11-21
> **Response to Reviewer 943i (3/3)**
>
> > Why PF3plat lacks behind Mast3R? Based on the experiments you show, your pose estimation is worse than Mast3R in almost all cases, and the NVS results are also worse than MVSplat in all scenarios (even on DL3DV in Table 3, where MVSplat was not trained on). One reasonable baseline to me is, get camera poses from Mast3R, and then run MVSplat directly. I wonder how your method compares to such a simple baseline?
>
> We wish to emphasize that MVSplat and Mast3R are included as references, and direct comparisons to them may be misleading due to fundamental differences in task setup and objectives. While pose estimation is undoubtedly important, the primary focus of our method is **pose-free novel view synthesis**.  Unlike Mast3R, which is specifically designed to estimate accurate camera poses by learning to find precise correspondences, leveraging extensive pretraining on large-scale datasets, our method solely leverages RGB images without relying on ground-truth pose or depth during either the training or inference phases.
>
> Nevertheless, it is worth noting that our approach can readily incorporate predictions from either RoMa or Mast3R. Below, we demonstrate that not only is our original performance on par with Mast3R, but also that by incorporating a more robust correspondence estimator, such as RoMa, our approach achieves superior performance compared to Mast3R:
>
> | Method       | PSNR   | SSIM  | LPIPS | Rot (Avg) | Rot (Med) | Trans (Avg) | Trans (Med) |
> |--------------|--------|-------|-------|-----------|-----------|-------------|-------------|
> | RoMa         | -      | -     | -     | 2.470     | 0.391     | 8.047       | 1.601       |
> | Mast3R       | -      | -     | -     | 2.555     | 0.751     | 9.775       | 2.830       |
> | Ours         | 22.347 | 0.763 | 0.205 | 1.965     | 0.751     | 10.113      | 4.785       |
> | Ours (RoMa)  | 23.121 | 0.798 | 0.191 | 1.874     | 0.723     | 7.674       | 3.891       |
>
>
> Moreover, we kindly direct the reviewer’s attention to Table 5(a), where our method demonstrates superior performance compared to Splatt3r, and Table 1, where it outperforms CoPoNeRF despite having lower pose estimation accuracy. This underscores the robustness of our approach to pose estimation noise. While achieving SOTA in every task is ideal, our primary contribution lies in addressing the pose-free feed-forward task, tackling challenges associated with 3DGS, and introducing a refinement module to enhance performance.
>
> Regarding the suggested simple baseline, we kindly refer the reviewer to the results provided in Table 4(0), which conveys a similar message to the one suggested by the reviewer. Furthermore, we demonstrate that each component we add results in noticeable improvements, culminating in our final performance surpassing the baseline.
>
> Finally, we would like to clarify that, as stated in L868, MVSplat was trained on DL3DV, contrary to the assumption that it was not. We can also leverage Mast3R's predictions; however, we believe this would be a repetitive experiment to the one conducted with RoMa, a current SOTA, and thus we leave it as future work.
>
> > In your ablation study table 4, what is the point of adding V, I-I, I-II, I-V, if they are just all N.A.? That is really weird to me. You can just describe them in texts.
>
> We wish to emphasize that our intention was to highlight the inherent difficulties of the pose-free feed-forward NVS task with 3D Gaussians, particularly the importance of coarse alignment and additional loss signals. While we understand that these details could alternatively be described in text, we chose to include them in the table to ensure clarity and accessibility for all readers. Presenting this information in a consolidated and visually accessible format reduces the risk of important details being overlooked and facilitates easier referencing during analysis. We believe this approach enhances the overall presentation of our results.
>
> > You use two paragraphs to motivate by mentioning the limitations of the previous methods. The real content for the coarse alignment is really just the third paragraph between line 186-191. The motivations part is actually unrelated to the coarse alignment but more on why your method is needed, so you should just put them in the introduction instead.
>
> We respectfully disagree with the observation that the first two paragraphs of Section 3.2.1 are unrelated to coarse alignment. These paragraphs are intended to highlight the necessity of coarse alignment, particularly in the context of our task with 3DGS in the pose-free feed-forward setup. The potential misalignments introduced during this process can lead to noisy gradients, which may significantly hinder learning. Coarse alignment is crucial to mitigate these issues and ensure the effectiveness of the overall framework. As such, we believe this motivation is directly tied to the content of Section 3.2.1 and is appropriately placed there.

---

> ### Author Response · Authors · 2024-11-25
>
> Dear Reviewer 943i,
>
> As the author-reviewer discussion period is coming to an end, we wanted to kindly remind you of our responses to your comments. We greatly value your feedback and are eager to address any additional questions or concerns you might have.
>
> Please let us know if there's any further information we can provide to facilitate the discussion process.
>
> We are highly appreciated for  your time and consideration.

---

> > ### Comment · Reviewer_943i · 2024-11-25
> >
> > Sorry for the slow response, and thanks for the detailed response to my questions, I appreciate your efforts!
> >
> > Many of my concerns are resolved, so I would increase my score to 5. However, the significance of the method is still questionable to me. Your method shows good performance over some previous pose-free work, but limited improvement over some other simple baselines like InstantSplat. Besides, I still have some points from your response below.
> >
> > 1. Pose refinements: thanks for your answer and it is clearer now. However, based on your Table 4 (I), it looks to me that removing depth refinement step actually even leads to improved performance in translation, while the rendering is almost the same as with depth refinement. This makes me wonder if this depth refinement step is really necessary.
> > 2. MVS Confidence: I checked your response and also your modified paper. If I understand correctly, the key difference to MVSplat is, that they use GT pose, while you use the estimated pose, but the core algorithm itself is basically the same, right?
> > 3. “MVSplat was trained on DL3DV”, this is not true for the original MVSplat (maybe it is true for your retrained version?)

---

> ### Author Response · Authors · 2024-11-25
>
> Thank you for your response!
>
> 0.    While we acknowledge that our method falls behind InstantSplat in certain aspects, we wish to highlight its apparent advantage over scene-specific optimization approaches: **inference time** (**Ours : 0.390s, InstantSplat 53s**) and **generalizability** . These are the  **common advantage shared by generalized methods**, since the very beginning of the generalized NVS task by pixelNeRF. Unlike scene-specific optimization methods, generalized NVS methods do not require such process, offering them advantages in real-world applications. Moreover, when equipped with test-time optimization (TTO), our method can achieve improved performance, **surpassing InstantSplat as shown in Tab.5(a)**, with significantly less time consumption. We hope this emphasizes the significance of our approach.
>
> 1.    We thank the reviewer for this observation. While it may appear from Table 4(I) that removing the depth refinement step improves translation performance, we would like to note that depth refinement contributes to other critical aspects, such as improving image quality (which is heavily influenced by the overall pose estimation accuracy in a pose-free setup) and ensuring robustness across varying scenarios. Moreover, the rendering metrics consistently improve as our refinement modules work **synergistically**. We believe depth refinement plays a vital role in the overall performance and stability of our framework, even if its isolated impact on certain metrics appears less significant.
>
> Additionally, we believe that in our setup, pose-free novel view synthesis (NVS) with 3D Gaussian Splatting (3DGS), both depth and camera pose are of paramount importance because the centers of the 3D Gaussians are determined by these two factors. While the translation scores may exhibit lower accuracy, we caution against concluding that the overall performance is degraded. The consistent improvements in rotation scores and image quality metrics suggest that any discrepancies in translation angles may have been compensated for, as demonstrated by the enhanced image quality.
>
> We also wish to highlight that the difference in PSNR scores for depth refinement is significant, especially considering that PSNR is measured on a logarithmic scale. This indicates that the performance improvement is substantial. Therefore, it would be inadvisable to conclude that the depth refinement module is unnecessary based solely on the translation angular difference. Our objective of accurate view rendering necessitates high precision in multiple factors, including depth, pose, confidence, and the learned 3D Gaussians. Each of these components contributes critically to the overall success of the method.
>
> 2.    Regarding the plane-sweeping cost volume construction, the reviewer's understanding is indeed correct! The additional difference lies in the guidance volume and how we aggregate it with the cost volume constructed with our predictions, and then the estimation of confidence scores, which we believe the reviewer has already identified.
>
> 3.    We apologize for the misunderstanding. What we meant was that the MVSplat scores reported in Table 3 (DL3DV) correspond to a retrained version specifically for DL3DV.
>
> Please let us know if our understanding is correct regarding 2 and 3.

---

> > ### Author Response · Authors · 2024-11-29
> >
> > Dear Reviewer 943i,
> >
> > We have included additional explanations in our responses, highlighting the differences between feed-forward approaches (ours) and per-scene optimization methods (InstantSplat), as well as further clarifying the importance of each of our proposed modules including the depth refinement module. As the author-reviewer discussion period is coming to an end, we would appreciate it if the reviewer could take a look at our responses and let us know if our responses have successfully addressed the reviewer’s concerns. We greatly value your feedback and are eager to address any further concerns or questions.

---

### Official Review · Reviewer_TPSu · 2024-11-04

**Soundness:** 3
**Presentation:** 3
**Contribution:** 3
**Rating:** 6
**Confidence:** 4

**Summary:**

This paper introduces PF3plat, a novel framework designed for novel view synthesis from unposed images in a single feed-forward pass. PF3plat leverages pre-trained monocular depth estimation and visual correspondence models to achieve an initial coarse alignment of 3D Gaussians. Subsequently, PF3plat incorporates refinement modules and geometry-aware scoring functions to further refine the depth and pose estimates derived from the coarse alignment to enhance the quality of view synthesis.

**Strengths:**

1. The task of novel view synthesis from unposed images in a single feed-forward pass is highly practical.
2. The paper demonstrates state-of-the-art results on Re10k and ACID, showcasing the effectiveness of the proposed method.
3.  The refinement modules designed in the paper have significantly improved the effectiveness.

**Weaknesses:**

1. PF3plat leverages a robust solver for pose estimation between each pair of cameras; thus, increasing the number of viewpoints significantly extends the feed-forward pass time.
2. PF3plat relies on the coarse alignment of 3D Gaussians, and a small overlap may affect the quality of the correspondence model.

**Questions:**

1. Why does using the pixel-wise depth offset estimation model promote consistency across views? (line 211)
2. How about the performance of PF3plat in dynamic scenes?

---

> ### Author Response · Authors · 2024-11-21
> **Response to Reviewer TPSu (1/2)**
>
> We highly appreciate the reviewer's comments and positive evaluation of our work! Our responses can be found below.
>
> > PF3plat leverages robust solver, which increases the inference time as the number of viewpoints increase.
>
> We appreciate the reviewer’s comment regarding the increase in feed-forward pass time with additional input views due to the exhaustive pair-wise pose estimation in our framework. We would like to emphasize that this limitation is common across other 3D reconstruction approaches, including SfM, Mast3R, and Dust3R, which similarly require comprehensive pair-wise pose estimation. Nevertheless, our method remains significantly faster than existing NeRF-based approaches, as shown in Table. 5 (b), highlighting its practicality in pose-free view synthesis tasks.
>
> > PF3plat relies on the coarse alignment of 3D Gaussians. Small overlapping may affect the quality of correspondence model.
>
> We acknowledge that cases with minimal overlap can present challenges for correspondence networks. However, compared to existing pose-free methods, such as DBARF, FlowCAM, Splatt3r, and CoPoNeRF, our method achieves the best performance, as shown in Tab. 1 and 3, highlighting its effectiveness and robustness.
>
>
> Moreover, we would like to highlight the flexibility of our approach, which enables seamless integration of various models, allowing us to select the most suitable one for specific scenarios. To enhance robustness in low-overlap situations, we can leverage more robust estimation models, such as RoMa (CVPR'24), to achieve better initial alignment. This can be further refined using our proposed methods and can also provide more accurate learning signals for the objective functions described in Sec. 3.3, which involve estimated correspondences. This adaptability underscores the strength of our design choices. To support this, we conducted an additional experiment, which is shown below:
>
> | Method       | PSNR   | SSIM  | LPIPS | Rot (Avg) | Rot (Med) | Trans (Avg) | Trans (Med) |
> |--------------|--------|-------|-------|-----------|-----------|-------------|-------------|
> | RoMa         | -      | -     | -     | 2.470     | 0.391     | 8.047       | 1.601       |
> | Ours         | 22.347 | 0.763 | 0.205 | 1.965     | 0.751     | 10.113      | 4.785       |
> | Ours (RoMa)  | 23.121 | 0.798 | 0.191 | 1.874     | 0.723     | 7.674       | 3.891       |
>
> From the results, we observe that our method significantly benefits from incorporating a more robust correspondence network, leading to notable improvements in both image quality metrics and pose accuracy. Note that the tendency for lower average errors but higher median errors compared to matching-based methods can be attributed to observations made in [1], where robust solvers tend to estimate precise poses. In contrast, learning-based approaches like ours produce more robust pose estimates, leading to greater consistency across diverse scenarios.
>
> [1] FAR: Flexible, Accurate and Robust 6DoF Relative Camera Pose Estimation, CVPR'24
>
> > Question: Why does using the pixel-wise depth offset estimation model promote consistency across views? (L211)
>
> In both monocular depth estimation [2, 3] and multi-view stereo literature [1], view synthesis has proven to be an effective supervisory signal for depth learning. As referenced in L212–L214, we cite Zhou et al., 2017, to highlight that supervision via view synthesis not only facilitates multi-view consistent depth estimation but also enhances accurate camera pose learning. This is particularly beneficial in our approach, where the centers of pixel-aligned 3D Gaussians are localized and determined using the estimated depth and camera pose. Accurate localization of each 3D Gaussian is critical for precise 3D reconstruction and view synthesis, which may explain why we empirically observed that estimating global scale and shift parameters resulted in lower performance.
>
> [1] DeepStereo: Learning to predict new views from the world’s imagery.
>
> [2] Unsupervised CNN for single view depth estimation: Geometry to the rescue.
>
> [3] Unsupervised monocular depth estimation with left-right consistency

---

> ### Author Response · Authors · 2024-11-21
> **Response to Reviewer TPSu (2/2)**
>
> > Question: How about the performance of PF3plat in dynamic scenes?
>
> We thank the reviewer for the thoughtful question. We acknowledge that PF3Plat may face challenges in dynamic scenes, which likely contributed to its performance on the ACID dataset, known for its dynamic elements. While our coarse alignment process, supported by monocular prediction and a robust solver, provides reliable depth and camera pose estimates, our current refinement module does not explicitly account for dynamic objects, which may affect accuracy.
>
> Addressing moving objects would require an additional module to explicitly model the deformation of 3D Gaussians, a feature that is not yet implemented in this work. This is an active area of research, with some works focusing on explicitly estimating deformation fields for each 3D Gaussian [1, 2, 3] or removing transient objects [4, 5, 6]. However, modeling 4D scenes in a single-feed-forward setup remains an unexplored challenge. We appreciate the reviewer’s suggestion, as this represents a natural and promising direction for future development with potential practical applications in fields such as robotics and egocentric vision.
>
> [1] 4D Gaussian Splatting for Real-Time Dynamic Scene Rendering, CVPR'24
>
> [2] Real-time Photorealistic Dynamic Scene Representation and Rendering with 4D Gaussian Splatting, ICLR'24
>
> [3] Gaussian-Flow: 4D Reconstruction with Dynamic 3D Gaussian Particle, CVPR'24
>
> [4] WildGaussians: 3D Gaussian Splatting in the Wild, NeurIPS'24
>
> [5] Gaussian in the Wild: 3D Gaussian Splatting for Unconstrained Image Collections, ECCV'24
>
> [6] EgoLifter: Open-world 3D Segmentation for Egocentric Perception, ECCV'24

---

> ### Author Response · Authors · 2024-11-25
>
> Dear Reviewer TPSu,
>
> As the author-reviewer discussion period is coming to an end, we wanted to kindly remind you of our responses to your comments. We greatly value your feedback and are eager to address any additional questions or concerns you might have.
>
> Please let us know if there's any further information we can provide to facilitate the discussion process.
>
> We are highly appreciated for  your time and consideration.

---

> > ### Comment · Reviewer_TPSu · 2024-11-25
> >
> > Thank you for the author's thorough responses. The responses have resolved my questions, and I will keep my positive rating.

---

> > > ### Author Response · Authors · 2024-11-25
> > >
> > > Thank you for your response! We’re delighted to hear that our reply addressed your concerns and appreciate your recognition of our work.

---

### Author Response · Authors · 2024-11-21
**General response**

We are grateful that the reviewers recognize the strengths of our work, including its well-written presentation (943i, RiDd), focus on an important topic, and proposal of a highly practical solution (TPSu, 943i, RiDd). Additionally, the reviewers acknowledged that our proposed modules are thoroughly ablated and proved to be highly effective, achieving state-of-the-art performance in all view synthesis benchmarks (TPSu, 943i, RiDd, HHva).

In our revised PDF, we have included additional figures detailing our architecture and qualitative comparisons for pose estimation. Furthermore, in this rebuttal, we have addressed key points raised by the reviewers, including:

- Inference time (TPSu),
- Coarse alignment (TPSu, RiDd),
- Extension to dynamic scenes (TPSu, HHva),
- Additional qualitative results (RiDd), and
- Cross-dataset generalization and further evaluation on DL3DV (HHva).

For reviewer 943i:
- we have clarified the missing details in the method section
- provided results of RoMa and PixelSplat as additional references, and
- justified why pose-required and matching-based methods should not be directly compared to pose-free methods like ours

We sincerely thank the reviewers for their constructive feedback and thorough evaluation. We hope that the revisions and clarifications provided in our rebuttal address all raised concerns, ensuring that our contributions are well-positioned within the scope of pose-free novel view synthesis.

---

### Author Response · Authors · 2024-12-02

Dear Reviewers,

We greatly appreciate your time and effort in reviewing our manuscript. As the author-reviewer discussion period is concluding, we would like to provide a final response that may assist in further considerations.

PF3plat, in which its main contribution lies in addressing the task of pose-free **novel view synthesis** in a single feed-forward pass using **3D Gaussian Splatting**, first adopts a novel approach that employs **coarse alignment** to tackle the unique challenges arising from the use of pixel-aligned 3D Gaussian Splatting. Specifically, misaligned 3D Gaussians across different views can induce noisy or sparse gradients, which destabilize training and hinder convergence, especially when common assumptions such as ground-truth camera poses, dense views, and substantial image overlaps are not met. We then
introduce **lightweight refinement modules and geometry-aware scoring functions**, which
not only enhance the reconstruction and view synthesis quality, but also prevent catastrophic forgetting issues typically associated with direct fine-tuning of coarse-alignment module, e.g., monocular depth. With this model, we evaluated on real-world large-scale datasets including **RealEstate10K, ACID, and DL3DV**, achieving state-of-the-art performance on all of them.

Regarding the remaining concerns that some reviewers have yet to respond to, we emphasized that while pose estimation is of prime importance ( yet our method achieves state-of-the-art or comparable results without relying on GT depth or correspondences, which previous works leverage for directly addressing pose estimation), our primary objective is **novel view synthesis**  (Reviewer HHva),  as clearly stated in the title, introduction, and contributions (L88).  It is also important to note that on top of Dust3R and Mast3R, methods exclusively for 3D reconstruction,  additional methods, such as NeRF or 3D Gaussian Splatting, is required to train a radiance field for novel view synthesis. This approach: 1) **is not a feature of VGGSfM or Dust3R**; 2) **requires a long optimization time for each scene**; and 3) **depends on GT camera poses, tracking, intrinsics, and depth for training**. Finally, our method can infer much faster than InstantSplat (Reviewer 943i), with InstantSplat taking 53 seconds and ours 0.390 seconds, highlighting the practicality and advantage of our single feed-forward approach, not to mention the **generalizability advantages that feed-forward approaches offer over optimization-based methods**. In terms of performance, we also show that incorporating test-time optimization enables our method to surpass InstantSplat as well as with significantly less time consumption.

We believe that our submission has sufficiently demonstrated its effectiveness through extensive experiments on real-world large-scale datasets for novel view synthesis, outperforming existing generalized pose-free NVS methods. We sincerely hope that the reviewers consider these for the subsequent discussion period.

---

### Meta-Review · Area_Chair_zzFu · 2024-12-17

**Metareview:**

This paper receives borderline final ratings of 6,5,6,5. The AC looked through the reviews, the rebuttal and the discussions between the reviewers and authors, and finally decided to reject the paper due to the remaining concerns raised by the two reviewers who gave negative ratings are valid. Specifically, there is a limited improvement over other simple baselines like instantsplat. The requirement of the depth refinement step is questionable, and the core algorithm is MVSplat with the replacement of estimated camera poses to replace the ground truth. Furthermore, the proposed method requires known camera intrinsics. In contrast, Vggsfm and Dust3R do not have this requirement. The results are also benchmarked on one public dataset, which is deemed insufficient. The two reviewers decided to give borderline rejects even after rebuttal and discussions. On the other hand, the two reviewers who are positive did not give very strong reasons to justify the acceptance of this work. The comments are this is a meaningful task, paper is well-written, highly practical, etc.

**Additional Comments On Reviewer Discussion:**

There are two reviewers who remain unconvinced after the rebuttal and discussions. They also brought up valid reasons to reject the paper.

---

### Decision · Program_Chairs · 2025-01-22

Reject